# The Blessing of Randomness: SDE Beats ODE in General Diffusion-based Image Editing

**Shen Nie**[1,2]*, **Hanzhong Allan Guo**[1,2], **Cheng Lu**[3], **Yuhao Zhou**[3],
**Chenyu Zheng**[1,2], **Chongxuan Li**[1,2]†

[1]Gaoling School of Artificial Intelligence, Renmin University of China, Beijing, China
[2]Beijing Key Laboratory of Big Data Management and Analysis Methods, Beijing, China
[3]Department of Computer Science and Technology, Tsinghua University, Beijing, China
`{nieshen, guohanzhong, chongxuanli}@ruc.edu.cn;`
`{lucheng.lc15, yuhaoz.cs, chenyu.zheng666}@gamil.com;`

## Abstract

We present a unified probabilistic formulation for diffusion-based image editing, where a latent variable is edited in a task-specific manner and generally deviates from the corresponding marginal distribution induced by the original stochastic or ordinary differential equation (SDE or ODE). Instead, it defines a corresponding SDE or ODE for editing. In the formulation, we prove that the Kullback-Leibler divergence between the marginal distributions of the two SDEs gradually decreases while that for the ODEs remains as the time approaches zero, which shows the promise of SDE in image editing. Inspired by it, we provide the SDE counterparts for widely used ODE baselines in various tasks including inpainting and image-to-image translation, where SDE shows a consistent and substantial improvement. Moreover, we propose *SDE-Drag* – a simple yet effective method built upon the SDE formulation for point-based content dragging. We build a challenging benchmark (termed *DragBench*) with open-set natural, art, and AI-generated images for evaluation. A user study on DragBench indicates that SDE-Drag significantly outperforms our ODE baseline, existing diffusion-based methods, and the renowned DragGAN. Our results demonstrate the superiority and versatility of SDE in image editing and push the boundary of diffusion-based editing methods. See the project page `https://ml-gsai.github.io/SDE-Drag-demo/` for the code and DragBench dataset.

## 1 Introduction

The primary objective of image editing tasks is to manipulate the content of a given image in a controlled manner without deviating from the data distribution. As a representative example, point-based dragging (Pan et al., 2023) provides a user-friendly way to manipulate the image content directly. Although the field has witnessed numerous endeavors exploring diverse editing methods (Meng et al., 2021; Rombach et al., 2022; Su et al., 2022; Wu & De la Torre, 2022; Couairon et al., 2022; Hertz et al., 2022; Zhao et al., 2022; Shi et al., 2023; Mou et al., 2023a), leveraging (large-scale) pre-trained diffusion models (Sohl-Dickstein et al., 2015; Ho et al., 2020; Song et al., 2020b) to achieve remarkable empirical performance, there remains an absence of genuine attempts to comprehend and expound upon this process from a probabilistic perspective.

To this end, we present a unified probabilistic formulation (see Fig. 1a, left panel) for diffusion-based image editing in Sec. 3.1, encompassing a wide range of existing work (see Tab. 1). In the formulation, the input image undergoes inversion or noise perturbation first, followed by manipulation or domain transformation to generate a task-specific latent variable. Starting from it, a stochastic differential equation (SDE) or a probability flow ordinary differential equation (ODE) is defined by a pretrained diffusion model. Notably, they differ from the ones used for sampling because the corresponding marginal distributions mismatch. In Sec. 3.2, we theoretically show the

---

*Work done during an internship at ShengShu, Beijing, China.
†Correspondence to Chongxuan Li.

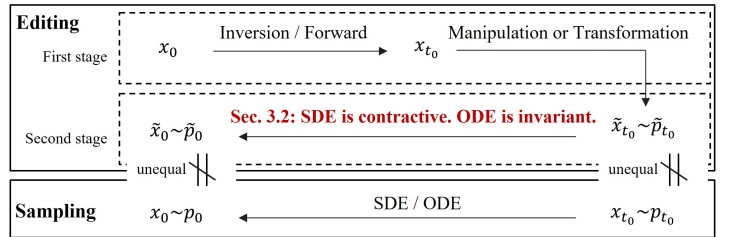
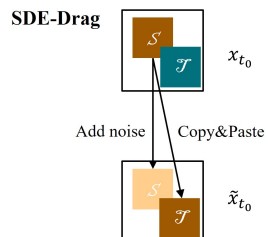

(a) **Technical contributions.** The left and right panels illustrate the unified probabilistic perspective for image editing in diffusion and how to manipulate the latent variable in SDE-Drag respectively. Best viewed in color.

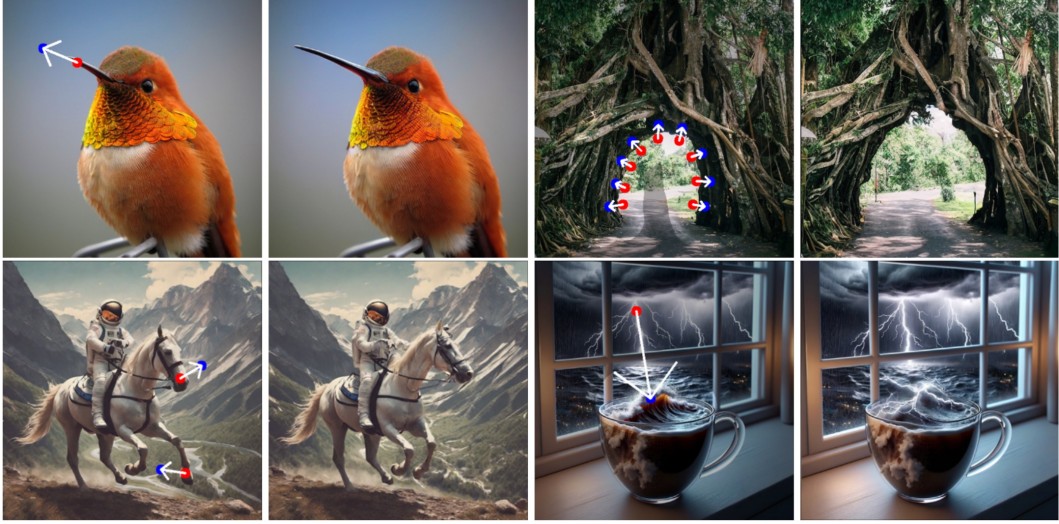

(b) **Results in Dragging.** The top and bottom rows visualize the results of SDE-Drag on real images and fake ones produced by Stable Diffusion (Rombach et al., 2022; Podell et al., 2023) and DALL·E 3 respectively. In particular, the last example shows that SDE-Drag can drag *miniature lightning* into the mug, which is mentioned in the text description yet missing in the original sample from DALL·E 3. See more in Appendix G.3.

Figure 1: **Overview of the paper.** (a) Technical contributions. (b) Visualization of SDE-Drag.

promise of the SDE formulation for general image editing: the Kullback-Leibler (KL) divergence between marginal distributions of the SDEs decreases as the time tends to zero while that of the ODEs remains the same.

Inspired by the analyses, we provide the SDE counterparts for widely used ODE baselines in various tasks. In particular, we propose a simple yet effective method (dubbed *SDE-Drag*) built upon the SDE formulation for point-based content dragging (Pan et al., 2023) in Sec. 4.1. Distinct from the prior work (Pan et al., 2023; Shi et al., 2023; Mou et al., 2023a), SDE-Drag manipulates the latent variable in a straightforward *copy-and-paste* manner (see Fig. 1a, right panel) instead of performing optimization in the latent space. We further build a challenging benchmark (termed *DragBench*) with 100 natural, art, and AI-generated samples open-set for evaluation in Sec. 4.2.

In Sec. 5, we conduct extensive experiments on various tasks including inpainting, image-to-image translation, and dragging, where the SDE counterparts show a consistent and substantial improvement over the widely used baselines. Notably, SDE-Drag can not only deal with open-set images but also improve the alignment between the prompt and sample from advanced AI-painting systems like Stable Diffusion and DALL·E 3 (see Fig. 1b).

## 2 BACKGROUND

We present preliminaries on diffusion models, SDE and ODE samplers, and data reconstruction algorithms in this section and discuss other related work in detail in Appendix C.

**Diffusion models.** Let $\boldsymbol{x}_0$ denote a $D$-dimensional random variable follows a data distribution $q_0(\boldsymbol{x}_0)$. Diffusion models (Sohl-Dickstein et al., 2015; Ho et al., 2020; Song et al., 2020b) introduce a forward process $\{\boldsymbol{x}_t\}_{t \in [0,T]}$ that gradually adds Gaussian noise to $\boldsymbol{x}_0$ and defines:

$$q_{0t}(\boldsymbol{x}_t|\boldsymbol{x}_0) = \mathcal{N}(\boldsymbol{x}_t|\sqrt{\alpha_t}\boldsymbol{x}_0, (1-\alpha_t)\boldsymbol{I}), \tag{1}$$

where $\alpha_t$ is a function of $t$ that decreases monotonically. Besides, $\alpha_0 = 1$ and $\alpha_T$ is sufficiently small to ensure $q_{0T}(\boldsymbol{x}_T|\boldsymbol{x}_0) \approx \mathcal{N}(0, \boldsymbol{I})$. We denote the distribution of $\boldsymbol{x}_t$ as $q_t(\boldsymbol{x}_t)$. It is equivalent to represent the forward process as the following *stochastic differential equation* (SDE):

$$d\boldsymbol{x}_t = f(t)\boldsymbol{x}_t dt + g(t)d\boldsymbol{w}_t, \tag{2}$$

where $\boldsymbol{w}_t$ is a standard Wiener process. For simplicity, we consider the VP type SDE (Song et al., 2020b), i.e., $f(t) = \frac{d \log \alpha_t}{2dt}$ and $g^2(t) = -\frac{d\alpha_t}{dt} - \frac{d \log \alpha_t}{dt}(1-\alpha_t)$. The forward process has an equivalent backward process referred to as the *reverse-time SDE*, which recovers $q_0(\boldsymbol{x}_0)$ from $q_T(\boldsymbol{x}_T)$:

$$d\boldsymbol{x}_t = \left[f(t)\boldsymbol{x}_t - g^2(t)\nabla_{\boldsymbol{x}_t} \log q_t(\boldsymbol{x}_t)\right] dt + g(t)d\bar{\boldsymbol{w}}_t, \tag{3}$$

where $\bar{\boldsymbol{w}}_t$ is a standard Wiener process with backward time and $\nabla_{\boldsymbol{x}_t} \log q_t(\boldsymbol{x}_t)$ is the score function of $q_t(\boldsymbol{x}_t)$. A diffusion model parameterizes the score functions by a neural network $\boldsymbol{s}_\theta(\boldsymbol{x}_t, t)$, which can be equivalently reparametrized as a noise prediction model $\boldsymbol{\epsilon}_\theta(\boldsymbol{x}_t, t)$. Taking $\boldsymbol{\epsilon}_\theta(\boldsymbol{x}_t, t)$ as an example, the reverse-time SDE is given by

$$d\boldsymbol{x}_t = \left[f(t)\boldsymbol{x}_t + \frac{g^2(t)}{\sqrt{1-\alpha_t}}\boldsymbol{\epsilon}_\theta(\boldsymbol{x}_t, t)\right] dt + g(t)d\bar{\boldsymbol{w}}_t. \tag{4}$$

We denote the marginal distribution of $\boldsymbol{x}_t$ defined by Eq. (4) as $p_t(\boldsymbol{x}_t)$ [1]. Insightfully, there exists a probability flow *ordinary differential equation* (ODE), whose marginal distribution for any given $t$ matches $p_t(\boldsymbol{x}_t)$ (Song et al., 2020b). Formally, the ODE is defined as:

$$d\boldsymbol{x}_t = \left[f(t)\boldsymbol{x}_t + \frac{g^2(t)}{2\sqrt{1-\alpha_t}}\boldsymbol{\epsilon}_\theta(\boldsymbol{x}_t, t)\right] dt. \tag{5}$$

**Samplers.** Samples can be generated from the diffusion model by discretizing the reverse SDE (i.e., Eq. (4)) (Ho et al., 2020; Bao et al., 2022b; Lu et al., 2022c) or ODE (i.e., Eq. (5)) (Song et al., 2020a; Lu et al., 2022b) from $T$ to 0. The ODE solvers often outperform SDE solvers given a few (e.g., $10-50$) steps and therefore are popular in sampling and editing. In particular, as a representative and widely adopted approach, DDIM (Song et al., 2020a) produces samples as follows:

$$\boldsymbol{x}_t = \sqrt{\alpha_t} \left(\frac{\boldsymbol{x}_s - \sqrt{1-\alpha_s}\boldsymbol{\epsilon}_\theta(\boldsymbol{x}_s, s)}{\sqrt{\alpha_s}}\right) + \sqrt{1-\alpha_t-\sigma_s^2}\boldsymbol{\epsilon}_\theta(\boldsymbol{x}_s, s) + \sigma_s \bar{\boldsymbol{w}}_s, \tag{6}$$

where $s > t$, $\sigma_s = \eta\sqrt{(1-\alpha_t)/(1-\alpha_s)}\sqrt{1-\alpha_s/\alpha_t}$, $\eta = 0$ and $\bar{\boldsymbol{w}}_s$ is a standard Gaussian noise. For fairness, we employ DDPM (Ho et al., 2020) as the SDE sampler in our experiment, which is also given by Eq. (6) with $\eta = 1$. We present the algorithms of both samplers in Appendix A.

**Data reconstruction.** To preserve relevant information about the input image, most editing methods require a way to produce a latent variable that can reconstruct the input. This can be easily implemented by ODE samplers because of its invertability. In fact, taking DDIM as an example, the latent variable can be directly produced by Eq. (6) with $s < t$ and $\eta = 0$, which is called *DDIM inversion*.

However, the SDE is nontrivial to invert because of the noise injected. The previous work (Wu & De la Torre, 2022) proposes to invert the forward process of DDPM as follows:

$$\boldsymbol{x}_s = \sqrt{\frac{\alpha_s}{\alpha_t}}\boldsymbol{x}_t + \sqrt{1-\frac{\alpha_s}{\alpha_t}}\boldsymbol{w}, \boldsymbol{w} \sim \mathcal{N}(0, \boldsymbol{I}), \tag{7}$$

$$\bar{\boldsymbol{w}}'_s = \frac{1}{\sigma_s}\left(\boldsymbol{x}_t - \sqrt{\alpha_t}\left(\frac{\boldsymbol{x}_s - \sqrt{1-\alpha_s}\boldsymbol{\epsilon}_\theta(\boldsymbol{x}_s, s)}{\sqrt{\alpha_s}}\right) - \sqrt{1-\alpha_t-\sigma_s^2}\boldsymbol{\epsilon}_\theta(\boldsymbol{x}_s, s)\right), \tag{8}$$

where $s > t$. Intuitively, it first adds noise to obtain latent variables from data in Eq. (7) and solves the "noise" $\bar{\boldsymbol{w}}'_s$ to be added to reconstruct the data through Eq. (6). It is easy to check that if we plug in $\bar{\boldsymbol{w}}'_s$ and $\boldsymbol{x}_s$ into the DDPM sampler in Eq. (6) with $\eta = 1$, we can reconstruct $\boldsymbol{x}_t$ perfectly. Note that it can be extended to all SDE solvers and we refer to the sampling process of SDE given $\bar{\boldsymbol{w}}'_s$ to *Cycle-SDE* for coherence. We present more details and algorithms of both reconstruction processes, as well as preliminary experiments on their reconstruction ability in Appendix A.

---

[1] We omit the dependence on $\theta$ in $p_t(\boldsymbol{x}_t)$ for simplicity.

Table 1: **Summary of prior work in a unified formulation.** I2I means the image-to-image translation task. As for the editing type, AM means to add a mask to combine the sampling result and noise input. CL and CP mean to change the label and prompt during sampling respectively. OL means to optimize the latent variable via gradient descent. See more details in Appendix B. In this paper, we add SDE counterparts for the ODE-based methods, as highlighted in blue in the last column.

| Task | Methods | $\boldsymbol{x}_0 \to \boldsymbol{x}_{t_0}$ | Edit | $\tilde{\boldsymbol{x}}_{t_0} \to \tilde{\boldsymbol{x}}_0$ | New results |
|---|---|---|---|---|---|
| Inpainting | Meng et al. (2021); Zhao et al. (2022) | Noise | AM | SDE | |
| Inpainting | Rombach et al. (2022) | Noise | AM | ODE | Tab. 2 |
| I2I | Couairon et al. (2022); Hertz et al. (2022) | ODE | CP | ODE | Fig. 2 |
| I2I | Su et al. (2022) | ODE | CL | ODE | Tab. 6 |
| I2I | Wu & De la Torre (2022) | Noise | CL | Cycle-SDE | |
| Dragging | Shi et al. (2023); Mou et al. (2023a) | ODE | OL | ODE | Fig. 3 |

## 3 THE BLESSING OF RANDOMNESS IN DIFFUSION-BASED IMAGE EDITING

In this section, we formulate a broad family of image-editing methods in diffusion models from a unified probabilistic perspective in Sec. 3.1 and show the potential benefits of SDE in Sec. 3.2.

### 3.1 A GENERAL PROBABILISTIC FORMULATION FOR IMAGE EDITING

In particular, given a pretrained diffusion model parameterized by $\boldsymbol{\theta}$, an image $\boldsymbol{x}_0$ to be edited, and a potential condition $c$ such as a label or text description, we identify two common stages for our general probabilistic formulation.

The first stage initially produces an intermediate latent variable $\boldsymbol{x}_{t_0}$ through either a noise-adding process (e.g., Eq. (7)) or an inversion process (e.g., Eq. (6) with $s < t$ and $\eta = 0$), where $t_0 \in (0, T]$ is a hyperparameter. Then, the latent variable $\boldsymbol{x}_{t_0}$ is manipulated manually or transferred to a different data domain by changing the condition $c$ in a task-specific manner[2]. We refer to the edited latent variable as $\tilde{\boldsymbol{x}}_{t_0}$, which deviates from the corresponding marginal distribution $p_{t_0}(\cdot)$ in general.

The second stage starts from $\tilde{\boldsymbol{x}}_{t_0}$ and produces the edited image $\tilde{\boldsymbol{x}}_0$ following either an ODE solver (e.g., Eq. (6) with $\eta = 0$), an SDE Solver (e.g., Eq. (6) with $\eta = 1$), or a Cycle-SDE process (e.g., Eq. (6) with $\eta = 1$ and $\bar{\boldsymbol{w}}'_s$ from Eq. (8)). This formulation encompasses a board family of methods, as summarized in Tab. 1. We illustrate these two stages in the left panel of Fig. 1a and please refer to Appendix B for more details.

Notably, the edited image $\tilde{\boldsymbol{x}}_0$ does not follow $p_0(\cdot)$ in general because the marginal distributions at time $t_0$ mismatch, making the editing process distinct from the well-studied sampling process. However, to the best of our knowledge, most of the prior work for image editing is empirical and there remains an absence of attempts to comprehend and expound upon the editing process precisely. Moreover, as mentioned in Sec. 2 and Tab. 1, due to the popularity of ODE in sampling and the property of easy to reverse, ODE is widely adopted in editing methods, and existing SDE-based approaches are rare and focus on a very specific task.

Our probabilistic perspective provides a general and principled way to study the editing process, from which we attempt to answer the following natural and fundamental problems in the paper. *Is there any guarantee on the sample quality of the edited image (i.e., the gap between its distribution and the data distribution)? Do the SDE and ODE formulations make any difference in editing?*

### 3.2 THEORETICAL ANALYSES

We present theoretical analyses here and we employ the widely adopted KL divergence to measure the closeness of distributions by default. Throughout the paper, we focus on analyzing the different

---

[2]Here we omit the detail that some existing methods manipulate the latent in multiple steps (see details in Appendix F). Nevertheless, our analyses in Sec. 3.2 apply to both the one-step and multiple-step editing.

behaviors of SDE and ODE formulations with mismatched prior distributions. Therefore, we assume the model characterizes the true score functions (see Sec. 6) for simplicity. In such a context, all of ODE, SDE, and Cycle-SDE can recover the data distribution without editing. Formally, we assume that they characterize the same marginal distribution $p_t$ and $p_t = q_t$ for any $t \in [0, T]$. We present a discussion in Appendix D.1 about the rationality of using $p$ as a surrogate for $q$ both theoretically and empirically. Besides, we consider general cases for all $0 \le s < t \le T$. By setting different values of $s$ and $t$, our results apply to both one-step and multiple-step editing. Please see Appendix D for more details and all proofs.

We first prove that the KL divergence between the marginal distributions of two SDEs with mismatched prior distributions gradually decreases while that for ODEs remains as the time approaches zero as shown by the following Theorems 3.1-3.2, respectively.

**Theorem 3.1** (Contraction of SDEs, see Appendix D.3). *Let $\tilde{p}_t$ and $p_t$ be the marginal distributions of two SDEs (see Eq. (4)) at time t. For any $0 \le s < t \le T$, if $\tilde{p}_t \ne p_t$, then under some mild regularity conditions listed in Lu et al. (2022a, Assumption A.1), it holds that*

$$D_{\mathrm{KL}}(\tilde{p}_s \| p_s) = D_{\mathrm{KL}}(\tilde{p}_t \| p_t) - \int_s^t g(\tau)^2 D_{\mathrm{Fisher}}(\tilde{p}_\tau \| p_\tau) d\tau < D_{\mathrm{KL}}(\tilde{p}_t \| p_t), \qquad (9)$$

*where $D_{\mathrm{KL}}(\cdot \| \cdot)$ denote the KL divergence and $D_{\mathrm{Fisher}}(\cdot \| \cdot)$ denote the Fisher divergence.*

**Theorem 3.2** (Invariance of ODEs, see Appendix D.3). *Let $\tilde{p}_t$ and $p_t$ be the marginal distributions of two ODEs (see Eq. (5)) at time t. For any $0 \le s < t \le T$, under some mild regularity conditions listed in Lu et al. (2022a, Assumption A.1), it holds that*

$$D_{\mathrm{KL}}(\tilde{p}_s \| p_s) = D_{\mathrm{KL}}(\tilde{p}_t \| p_t). \qquad (10)$$

Theorems 3.1-3.2 share the same spirit as Lyu (2012); Lu et al. (2022a). In particular, they follow from the Fokker-Plank equation and a detailed computation of the time derivative of $D_{\mathrm{KL}}(\tilde{p}_t \| p_t)$.

Moreover, with a stronger yet standard assumption (i.e., the log-Sobolev inequality holds for the data distribution), we can obtain a linear convergence rate in the SDE formulation. Namely, we have $D_{\mathrm{KL}}(\tilde{p}_s \| p_s) = \exp(O(-(t-s))) D_{\mathrm{KL}}(\tilde{p}_t \| p_t)$. We present the formal statement and more details in Appendix D.4 for completeness. Further, we conduct a toy experiment on a one-dimensional Gaussian mixture data where the true score function has an analytic form and the log-Sobolev inequality holds to illustrate the above results clearer in Appendix E.

The analysis of Cycle-SDE is more difficult because the random variables $\bar{w}'_s$ in the sampling path are correlated. Therefore, it is highly nontrivial to obtain a quantitative convergence rate for Cycle-SDE. However, we can still prove a similar "contractive" result as in SDE using the data processing inequality for KL divergence (Cover, 1999). The result is presented in the following theorem.

**Theorem 3.3** (Contraction of Cycle-SDEs, see Appendix D.5). *Let $p_t$ and $\tilde{p}_t$ be the marginal distributions of two Cycle-SDEs (e.g., see Eq. (6) with $\eta = 1$ and $\bar{w}'_s$ from Eq. (8)) at time t, respectively. For any $0 \le s < t \le T$, if $p_t \ne \tilde{p}_t$, then*

$$D_{\mathrm{KL}}(\tilde{p}_s \| p_s) < D_{\mathrm{KL}}(\tilde{p}_t \| p_t). \qquad (11)$$

In summary, we show that the additional noise in the SDE formulation (including both the original SDE and Cycle-SDE) provides a way to reduce the gap caused by mismatched prior distributions, while the gap remains invariant in the ODE formulation, suggesting the blessing of randomness in diffusion-based image editing. Inspired by the theory, we propose the SDE counterparts for representative ODE baselines across various editing tasks (as highlighted in blue in Tab. 1) and show a consistent and substantial improvement in all settings (see Sec. 5).

## 4 DRAG YOUR DIFFUSION IN THE SDE FORMULATION

Among various image editing tasks, point-based dragging (Pan et al., 2023) provides a user-friendly way to manipulate the image content directly and has attracted more and more attention in diffusion models recently (Shi et al., 2023; Mou et al., 2023a). Inspired by the formulation and theory in Sec. 3, we investigate how to drag images in the SDE formulation (see Sec. 4.1) and introduce a challenging benchmark with 100 open-set images for evaluation in Sec. 4.2.

---

**Algorithm 1** SDE-Drag

---

**Require:** $x_0, a_s, a_t$; hyper-parameters: $r, m, n, t_0 \in (0, T), \alpha \in (1, +\infty), \beta \in [0, 1]$
    **for** $j$ in $\{1, 2 \ldots, m\}$ **do**
        Obtain $x_{t_0}, \{\bar{w}'_i\}_{i=1}^n$ according to Eq. (7-8) with $x_0, n$ and $t_0$    ▶ Add and memorize noise
        Obtain $\tilde{x}_{t_0}$ according to Eq. (12) with $\alpha, \beta, r$ and $x_{t_0}$    ▶ Manipulate the latent variable
        Obtain $\tilde{x}_0$ according to Eq. (6) with $\eta = 1, \{\bar{w}'_i\}_{i=1}^n$ and $\tilde{x}_{t_0}$    ▶ Sample with Cycle-SDE
        $x_0 \leftarrow \tilde{x}_0$    ▶ Update the image
    **end for**
    **Return** $x_0$

---

## 4.1 SDE-DRAG

We propose a simple yet effective dragging algorithm based on the SDE formulation, dubbed *SDE-Drag*. We present SDE-Drag following the unified formulation in Sec. 3.

In the first stage, we first add noise to obtain the intermediate representation $x_{t_0}$ and obtain a series of "noise" $\bar{w}'_s$ according to Eqs. (7-8). Distinct from the prior work (Pan et al., 2023; Shi et al., 2023; Mou et al., 2023a), we manipulate the latent variable in a straightforward *copy-and-paste* manner (see Fig. 1a, right panel) instead of performing optimization in the latent space. In particular, given a source point $a_s$ and a target point $a_t$ provided by the user, let $\mathcal{S}$ and $\mathcal{T}$ denote the two squares with side length $2r$ centered at the two points, respectively, where $r$ is a hyperparameter. Then, we first make a copy of $x_{t_0}$ as $\tilde{x}_{t_0}$ and modify $\tilde{x}_{t_0}$ as follows:

$$\tilde{x}_{t_0}[\mathcal{T}] = \alpha x_{t_0}[\mathcal{S}], \tilde{x}_{t_0}[\mathcal{S} \setminus \mathcal{T}] = (\beta x_{t_0} + \sqrt{1 - \beta^2}\epsilon)[\mathcal{S} \setminus \mathcal{T}], \epsilon \sim \mathcal{N}(0, I), \quad (12)$$

where $\alpha \in (1, \infty), \beta \in [0, 1)$ are hyperparameters, $x[\mathcal{S}]$ takes pixels in $x$ that corresponds to the set $\mathcal{S}$ and $\setminus$ is the complementary operation for sets. Intuitively, SDE-Drag directly copies the noisy pixels from the source to the target and amplifies them by a factor $\alpha$. Besides, SDE-Drag perturbs the source with a Gaussian noise by a factor $\beta$ to avoid duplicating the content.

In the second stage, we directly produce $\tilde{x}_0$ by a Cycle-SDE process defined by Eq. (6) with $\eta = 1$, $\bar{w}_s = \bar{w}'_s$ and $x_{t_0} = \tilde{x}_{t_0}$. Besides, when an optional binary mask is provided, the unmasked area of the latent representation $\tilde{x}_t$ at every step $t$ is replaced by the corresponding unmasked area of the latent representation $x_t$[3] in the noise-adding process. To our knowledge, this is the first paper to investigate SDE formulations in dragging among prior work (Shi et al., 2023; Mou et al., 2023a).

When the target point is far from the source point, it is challenging to drag the content in a single operation (Pan et al., 2023). To this end, we divide the process of Drag-SDE into $m$ steps along the segment joining the two points and each one moves an equal distance sequentially. We present the whole process in Algorihtm 1 with $n$ discretization steps in sampling. $m$ and $n$ are hyperparamters.

We emphasize that we employ default values of all hyperparameters (e.g., $\alpha, \beta, m$, and $n$) for all input images and SDE-Drag is not sensitive to the hyperparameters, as detailed in Sec. 5.3. Moreover, it is easy to extend SDE-Drag by adding multiple source-target pairs either simultaneously or sequentially. For more implementation details, please refer to Appendix F.6. Our experiments (see Sec. 5.3) on a benchmark introduced later demonstrate the effectiveness of the SDE-Drag over an ODE baseline and representative prior work (Pan et al., 2023; Shi et al., 2023).

## 4.2 DRAGBENCH

Built upon large-scale text-to-image diffusion models (Rombach et al., 2022), SDE-Drag and ODE-based diffusion (Shi et al., 2023; Mou et al., 2023a) can potentially deal with open-set images, which is desirable. However, existing benchmarks (Pan et al., 2023) are restricted to specific domains like human faces. To this end, we introduce *DragBench*, a challenging benchmark consisting of 100 image-caption pairs from the internet that cover animals, plants, art, landscapes, and high-quality AI-generated images. Each image has one or multiple pairs of source and target points and some images are associated with binary masks. Please refer to Appendix F.4 for more details.

---

[3]Note that our analyses in Sec. 3.2 apply to any time interval including this case.

Table 2: **Results in inpainting.** Inpaint-ODE and Inpaint-SDE employ DDIM (Song et al., 2020a) and DDPM (Ho et al., 2020) respectively. Inpaint-SDE outperforms Inpaint-ODE in all settings.

| | Small Mask (average mask ratio 0.2) | | | | | | Large Mask (average mask ratio 0.4) | | | | | |
| | FID ↓ | | | LPIPS ↓ | | | FID ↓ | | | LPIPS ↓ | | |
| # steps | 25 | 50 | 100 | 25 | 50 | 100 | 25 | 50 | 100 | 25 | 50 | 100 |
|---|---|---|---|---|---|---|---|---|---|---|---|---|
| Inpaint-ODE | 5.84 | 5.86 | 5.86 | 0.227 | 0.227 | 0.227 | 17.73 | 17.57 | 17.50 | 0.363 | 0.361 | 0.361 |
| Inpaint-SDE | **5.18** | **4.97** | **4.91** | **0.218** | **0.215** | **0.215** | **16.43** | **15.48** | **15.27** | **0.351** | **0.345** | **0.343** |

## 5 EXPERIMENT

We conduct experiments on various tasks including inpainting (see Sec. 5.1), image-to-image (I2I) translation (see Sec. 5.2), and dragging (see Sec. 5.3), where the SDE counterparts show a consistent and substantial improvement over the widely used baselines. We also analyze the sensitivity of hyperparameters and time efficiency (see Sec. 5.4).

### 5.1 INPAINTING

**Setup.** Inpainting aims to complete missing values based on observation. For broader interests, we follow the LDM (Rombach et al., 2022) to employ the Places (Zhou et al., 2017) dataset for evaluation and adopt FID (Heusel et al., 2017) and LPIPS (Zhang et al., 2018) to measure the sample quality. For simplicity and fairness, the base model (i.e., Stable Diffusion 1.5 (Rombach et al., 2022) trained on LAIOB-5B (Schuhmann et al., 2022)), and most of the hyperparameters remain the same as the ODE baseline (Rombach et al., 2022). The only tuned hyperparameter is the number of sampling steps $n$ and we conduct systematical experiments with 25, 50, and 100 steps. Please refer to Appendix F.1 for more details.

**Results.** As presented in Tab. 2, the SDE counterpart significantly outperforms the ODE baseline across all settings under both the FID and PLIPS metrics. Notably, with a few steps (e.g., 25), where ODE is widely recognized to have a significant advantage in normal sampling, SDE still manages to achieve superior results. Further, the performance of SDE at 25 steps surpasses that of ODE at 100 steps, which strongly suggests the usage of SDE in inpainting. Additionally, we present the results of high-order algorithms in Tab. 5 for completeness and the conclusion remains.

### 5.2 IMAGE-TO-IMAGE TRANSLATION

**Setup.** As visualized in Fig. 14 in Appendix I.2, I2I aims to transfer the input images to another domain by changing its label or text description. We provide the SDE counterparts for two representative ODE approaches. Due to space limit, we present the results on DiffEdit (Couairon et al., 2022) here and those on DDIB (Su et al., 2022) in Appendix G.2. For fairness, we use Stable Diffusion 1.5 (Rombach et al., 2022) trained on LAION-5B (Schuhmann et al., 2022)) and employ image-caption pairs from the COCO dataset (Lin et al., 2014) and annotations from the COCO-BISON dataset (Hu et al., 2019) following the DiffEdit. Similarly, we adopt the same evaluation metrics including FID, LPIPS, and CLIP-Score (Radford et al., 2021). which measures the alignment between edited images and target prompts. As suggested by DiffEdit, we tune a single key hyperparameter, i.e. the time index of the latent variable $t_0$, and keep others unchanged. The SDE counterpart (termed DiffEdit-SDE) employs the Cycle-SDE. Please refer to Appendix F.3 for more details.

**Results.** Quantitatively, DiffEdit-SDE outperforms the ODE baseline (termed DiffEdit-ODE) under all metrics with a wide range of $t_0$, as presented in Fig. 2. Qualitatively, as visualized in Fig. 14, DiffEdit-SDE achieves a better image-text alignment and higher image fidelity simultaneously. We observe similar results in experiments with DDIB in Appendix G.2.

### 5.3 DRAGGING

**Setup.** As visualized in Fig. 17 in Appendix I.4, dragging (Pan et al., 2023) aims to drag the content following the instructions of the source points and target points. We compare our SDE-Drag against

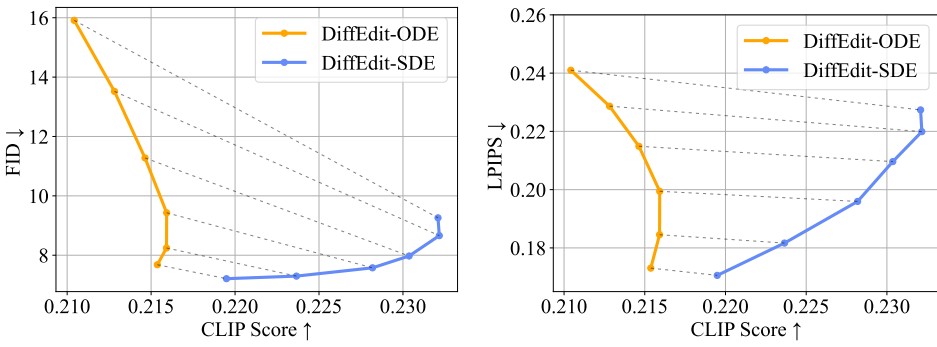

(a) Trade-offs between FID and CLIP-Score  (b) Trade-offs between LPIPS and CLIP-Score

Figure 2: **Results in I2I (DiffEdit).** We consider $t_0 \in \{0.3, 0.4, 0.5, 0.6, 0.7, 0.8\}$. With the same value of $t_0$ (linked by dashed lines), DiffEdit-SDE outperforms DiffEdit-ODE under all metrics.

a direct ODE baseline (termed ODE-Drag), an ODE-based prior work (Shi et al., 2023), and the DragGAN (Pan et al., 2023). We focus on the challenging and desirable dragging task on open-set images and there is no standard benchmark in the literature as far as we know. Therefore, we conduct a user study on the DragBench introduced in Sec. 4.2, as summarized below.

**User Study.** There were 6 participants and 100 questions for each one vs. one (e.g., SDE-Drag vs. ODE-Drag) comparison by default. There were 22 questions to compare with DragGAN because it is restricted to specific domains such as cats and lions. In each question, participants were presented with original images paired with two differently edited images produced by distinct models. Participants were tasked with selecting a better image between the pair, which is known as the Two-Alternative Forced Choice methodology commonly used in the literature (Kawar et al., 2023; Bar-Tal et al., 2022; Kolkin et al., 2019; Park et al., 2020). See more details in Appendix F.5.

**Implementation.** Throughout the paper, the hyperparameters in SDE-Drag are $r = 5$, $n = 120$, $t_0 = 0.6T$, $\alpha = 1.1$, $\beta = 0.3$ and $m = \lceil \|a_s - a_t\|/2 \rceil$, where $\|a_s - a_t\|$ denotes the Euclidean distance between $a_s$ and $a_t$. If multiple points are present, $m$ is determined based on the greatest distance among all pairs of points. To enjoy relatively high classifier-free guidance (CFG) and numerical stability simultaneously, we linearly increase the CFG from 1 to 3 as the time goes from 0 to $t_0$. Note that SDE-Drag is insensitive to most of the hyperparameters, as detailed in Sec. 5.4.

ODE-Drag shares all the hyperparameters as SDE-Drag except that it employs ODE inversion and sampling algorithms. Besides, following Shi et al. (2023), we integrate an optional LoRA (Hu et al., 2021) finetuning process for all diffusion-based methods. Nevertheless, SDE-Drag without LoRA still outperforms ODE-Drag and DragGAN. See more implementation details in Appendix F.6.

**Results.** As shown in Fig. 3, SDE-Drag significantly outperforms all competitors including the direct baseline ODE-Drag, representative ODE-based prior work (Shi et al., 2023) and DragGAN (Pan et al., 2023) on the challenging DragBench through a comprehend user study. For completeness, we also present the results without LoRA against ODE-Drag and DragGAN in Fig. 7 in Appendix G.3 and the conclusion remains. In addition, the visualization results in Appendix I.3 show that SDE-Drag can produce high-quality images edited in a desired manner, which agrees with the user study. All these results together with the ones in inpainting (Sec. 5.1) and I2I (Sec. 5.2) clearly demonstrate the superiority and versatility of SDE in image editing.

We further highlight that SDE-Drag is able to deal with multiple points simultaneously or sequentially on open-domain images and improve the alignment between the prompt and sample from the advanced AI-painting systems like DALL·E 3 and Stable Diffusion (Podell et al., 2023) in Figs. 18-20, advancing the area of interactive image generation.

## 5.4 SENSITIVITY ANALYSIS AND TIME EFFICIENCY

We perform a sensitivity analysis for all experiments. Overall, we did not tune the hyperparameters heavily and the SDE-based methods are not sensitive to most of the hyperparameters. In particular,

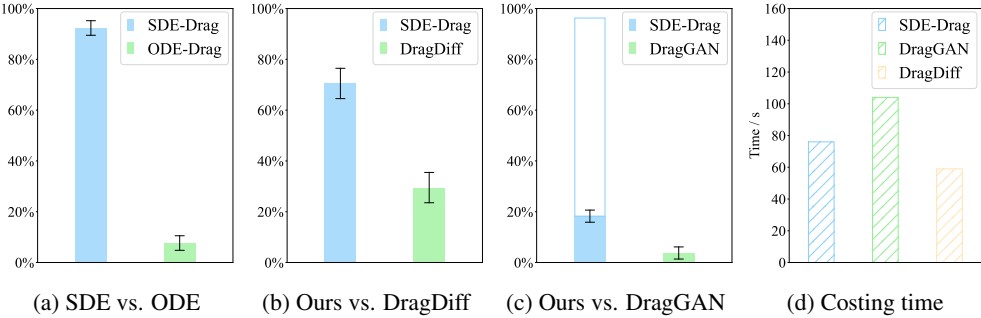

(a) SDE vs. ODE     (b) Ours vs. DragDiff     (c) Ours vs. DragGAN     (d) Costing time

Figure 3: **Results in dragging**. (a-c) present the preference rates (with $95\%$ confidence intervals) of SDE-Drag over ODE-Drag, DragDiffusion, and DragGAN. SDE-Drag significantly outperforms all competitors. The blank box in (c) denotes the ratio of the open-domain images in DragBench that DragGAN cannot edit. (d) shows that the average time cost per image is comparable for all methods.

Tab. 2 and Fig. 2 suggest that SDE-based methods can achieve good results in a wide range of key hyperparameters in inpainting and I2I. Below we focus on the analysis of SDE-Drag where we perform preliminary experiments on several images over a small set of values for hyperparameters.

**Time $t_0$ and amplification factor $\alpha$.** We evaluated SDE-Drag with $t_0/T \in \{0.4, 0.5, 0.6, 0.7, 0.8\}$. As identified in Meng et al. (2021) and confirmed in Fig. 9, a higher $t_0$ improves the fidelity while lowering the faithfulness of the edited images and vice versa. However, the overall performance is similar for values between $0.5T$ and $0.7T$ and we set $t_0 = 0.6T$ by default. We also evaluate SDE-Drag with $\alpha \in \{1.0, 1.1, 1.2\}$ and the default value of $1.1$ performs slightly better (see Fig. 10).

**Perturbation factor $\beta$.** We evaluate SDE-Drag with $\beta \in \{0.1, 0.2, 0.3, 0.4, 0.5, 1\}$. Intuitively, when $\beta = 1$, the signal at the source point is retained, leading to "copying" the content. For typical dragging, SDE-Drag is robust to values between $0.1$ and $0.5$. Fig. 11 visualizes the effect of $\beta$.

**Dragging steps $m$.** As discussed in Sec. 4.1, when the target point is far from the source point, it is challenging to drag the content with $m = 1$. As illustrated in Fig. 12, it is sufficient to use an adaptive strategy with $m = \lceil \|a_s - a_t\|/2 \rceil$. SDE-Drag is robust when using a factor between $1/4$ and $1$ in the adaptive strategy.

**LoRA.** We perform a systematical ablation study of LoRA in terms of user preference. We provide visualization results in Fig. 8 and more detailed analyses in Appendix F.6.

Notably, although the optimal hyperparameters may vary for each image, SDE-Drag works well in a wide range of hyperparameters and we provide a set of default hyperparameters for all images.

We now discuss the time efficiency. Overall, SDE-based methods achieve excellent results without increasing the computational time. As shown in Tab. 7 in Appendix G.4, the SDE counterpart takes nearly the same time as the direct ODE baseline in all experiments. Besides, SDE-Drag, DragDiffusion, and DragGAN have similar time efficiency as well, as shown in Figure 3d.

## 6   CONCLUSION AND DISCUSSION

We present a unified probabilistic formulation for diffusion-based image editing encompassing a wide range of existing work. We theoretically show the promise of the SDE formulation for general image editing. We propose a simple yet effective dragging algorithm (SDE-Drag) based on the SDE formulation and a challenging benchmark with 100 open-set images for evaluation. Our results in inpainting, image-to-image translation, and dragging clearly demonstrate the superiority and versatility of SDE in general image editing.

There are several limitations. Theoretically, we do not consider the model approximation error and discretization error, which can potentially be addressed based on existing work (Lu et al., 2022a; Lee et al., 2022; 2023; Chen et al., 2022; 2023). Besides, it is nontrivial to obtain convergence rate results for Cycle-SDE, which requires task-specific assumptions and analysis. Empirically, we observe that the open-set dragging problem is far from solved and we present failure cases in Appendix J.

**Ethics Statement.** This paper enhances the effectiveness of various image editing techniques, which leads to numerous societal benefits, including boosting productivity in visual industries, introducing modern tools to educational curricula, and democratizing digital content creation for a wider audience. However, it can also be harnessed to produce deceptive images or "deepfakes", and may give rise to ethical concerns around image alterations that misrepresent reality or history if not used properly. Besides, the editing process relies on a pretrained diffusion model. If the model is trained on biased data, the editing process may amplify the bias.

## ACKNOWLEDGEMENTS

This work was supported by NSF of China (No. 62076145); Beijing Outstanding Young Scientist Program (No. BJJWZYJH012019100020098); Major Innovation & Planning Interdisciplinary Platform for the "Double-First Class" Initiative, Renmin University of China; the Fundamental Research Funds for the Central Universities, and the Research Funds of Renmin University of China (No. 22XNKJ13). C. Li was also sponsored by Beijing Nova Program (No. 20220484044).

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

---

**Algorithm 2** DDIM sampler

---

**Require:** $x_T$
1: **for** $i = n + 1, \ldots, 2$ **do**
2: $\quad x_{t_{i-1}} = \sqrt{\alpha_{t_{i-1}}} \left( \frac{x_{t_i} - \sqrt{1 - \alpha_{t_i}} \boldsymbol{\epsilon}_\theta(x_{t_i}, t_i)}{\sqrt{\alpha_{t_i}}} \right) + \sqrt{1 - \alpha_{t_{i-1}}} \boldsymbol{\epsilon}_\theta(x_{t_i}, t_i)$
3: **end for**
4: **return** $x_0$

---

**Algorithm 3** DDPM sampler

---

**Require:** $x_T$
1: **for** $i = n + 1, \ldots, 2$ **do**
2: $\quad \sigma_{t_i} = \sqrt{(1 - \alpha_{t_{i-1}})/(1 - \alpha_{t_i})} \sqrt{1 - \alpha_{t_i}/\alpha_{t_{i-1}}}$
3: $\quad \bar{w}_i \sim \mathcal{N}(0, \boldsymbol{I})$
4: $\quad x_{t_{i-1}} = \sqrt{\alpha_{t_{i-1}}} \left( \frac{x_{t_i} - \sqrt{1 - \alpha_{t_i}} \boldsymbol{\epsilon}_\theta(x_{t_i}, t_i)}{\sqrt{\alpha_{t_i}}} \right) + \sqrt{1 - \alpha_{t_{i-1}} - \sigma_{t_i}^2} \boldsymbol{\epsilon}_\theta(x_{t_i}, t_i) + \sigma_{t_i} \bar{w}_i$
5: **end for**
6: **return** $x_0$

---

# A  BACKGROUND IN DETIAL

In this section, we detail the sampling algorithm (see Appendix A.1) and the data reconstruction methods (see Appendix A.2) employed in our experiments. In addition, we present an empirical study of the reconstruction capability of Cycle-SDE and DDIM inversion in Appendix A.3.

## A.1  SAMPLERS

We denote the noise prediction network as $\boldsymbol{\epsilon}_\theta(x_t, t)$ and define a time sequence $\{t_i\}_{i=1}^{n+1}$ increasing from $t_1 = 0$ and $t_{n+1} = T$, where $n$ is the number of sampling steps. Specifically, when classifier-free guidance (Ho & Salimans, 2022) is employed, we have $\boldsymbol{\epsilon}_\theta(x_t, t) = \boldsymbol{\epsilon}_\theta(x_t, t, \varnothing) + s(\boldsymbol{\epsilon}_\theta(x_t, t, c) - \boldsymbol{\epsilon}_\theta(x_t, t, \varnothing))$, where $s$ is the guidance scale, $c$ is the conditional embedding and $\varnothing$ is the unconditional embedding. We detail two representative ODE and SDE samplers, i.e., DDIM (Song et al., 2020a) and DDPM (Ho et al., 2020) in Algorithm 2 and Algorithm 3, respectively.

## A.2  DATA RECONSTRUCTION

With the same notation as Appendix A.1, below we summarize the data reconstruction methods based on ODE or SDE solver.

Due to the invertibility of ODE, given $x_0$, we can deduce the latent representation $x_T$ that ensures the reconstruction of $x_0$ via ODE sampling. Based on Eq. (5), the general formulation of ODE inversion is defined as:

$$x_T = x_0 + \int_0^T \left[ f(t)x_t + \frac{g^2(t)}{2\sqrt{1 - \alpha_t}} \boldsymbol{\epsilon}_\theta(x_t, t) \right] dt, \tag{13}$$

we detail the widely used DDIM inversion in Algorithm 4, which is a special discretization method of general ODE inversion.

In the context of an SDE solver, given $x_0$, we first log a forward trajectory $\{x_{t_i}\}_{i=1}^{n+1}$ employing the forward precoss (i.e., Eq. (7)). Recall that each sampling step (e.g., Eq. (6) with $\eta = 1$) of the SDE solver (first order) can be denoted as $x_{t_{i-1}} = \boldsymbol{f}(\boldsymbol{\epsilon}_\theta, x_{t_i}, t_i) + \sigma_{t_i} \bar{w}_{t_i}$, where $\boldsymbol{f}$ is a function defined by sampling algorithm. Consequently, we can analyticly compute $\bar{w}_{t_i} = (\boldsymbol{f}(\boldsymbol{\epsilon}_\theta, x_{t_i}, t_i) - x_{t_{i-1}})/\sigma_{t_i}$ based on $x_{t_i}$ and $x_{t_{i-1}}$ logged in the forward trajectory. With all $\{w_{t_i}\}_{i=2}^{n+1}$ calculated, we can reconstruct $x_0$ employ the SDE solver sampling from $x_T$ (i.e., $x_{t_{n+1}}$). We call these procedures Cycle-SDE and describe a DDPM-based Cycle-SDE in Algorithm 5.

---

**Algorithm 4** DDIM inversion

---

**Require:** $x_0$
1: **for** $i = 2, \ldots n + 1$ **do**
2: $\quad x_{t_i} = \sqrt{\alpha_{t_i}} \left( \frac{x_{t_{i-1}} - \sqrt{1 - \alpha_{t_{i-1}}} \, \epsilon_\theta(x_{t_{i-1}}, t_{i-1})}{\sqrt{\alpha_{t_{i-1}}}} \right) + \sqrt{1 - \alpha_{t_i}} \, \epsilon_\theta(x_{t_{i-1}}, t_{i-1})$
3: **end for**
4: **return** $x_T$

---

**Algorithm 5** Cycle-SDE (based on DDPM)

---

**Require:** $x_0$
1: **for** $i = 2, \ldots n + 1$ **do**
2: $\quad x_{t_i} = \sqrt{\alpha_{t_i} / \alpha_{t_{i-1}}} \, x_{t_{i-1}} + \sqrt{1 - \alpha_{t_i} / \alpha_{t_{i-1}}} \, w, \, w \sim \mathcal{N}(0, I)$
3: $\quad \sigma_{t_i} = \sqrt{(1 - \alpha_{t_{i-1}})/(1 - \alpha_{t_i})} \sqrt{1 - \alpha_{t_i} / \alpha_{t_{i-1}}}$
4: $\quad \bar{w}_i = \frac{1}{\sigma_{t_i}} \left( x_{t_{i-1}} - \sqrt{\alpha_{t_{i-1}}} \left( \frac{x_{t_i} - \sqrt{1 - \alpha_{t_i}} \, \epsilon_\theta(x_{t_i}, t_i)}{\sqrt{\alpha_{t_i}}} \right) - \sqrt{1 - \alpha_{t_{i-1}} - \sigma_{t_i}^2} \, \epsilon_\theta(x_{t_i}, t_i) \right)$
5: $\quad$ Record $\sigma_{t_i}$ and $\bar{w}_i$.
6: **end for**
7: **for** $i = n + 1, \ldots 2$ **do**
8: $\quad x_{t_{i-1}} = \sqrt{\alpha_{t_{i-1}}} \left( \frac{x_{t_i} - \sqrt{1 - \alpha_{t_i}} \, \epsilon_\theta(x_{t_i}, t_i)}{\sqrt{\alpha_{t_i}}} \right) + \sqrt{1 - \alpha_{t_{i-1}} - \sigma_{t_i}^2} \, \epsilon_\theta(x_{t_i}, t_i) + \sigma_{t_i} \bar{w}_i$
9: **end for**
10: **return** $x_T, \{\bar{w}_i\}_{i=1}^n$

---

## A.3 EMPIRICAL STUDY OF CYCLE-SDE

In this section, we discuss the reconstruction capability of Cycle-SDE with classifier-free guidance (CFG) and compare it to the widely adopted ODE inversion method in image reconstruction.

In the procedure of Cycle-SDE, we first log a forward trajectory $\{x_{t_i}\}_{i=1}^{n+1}$ via the forward process, and the sampling process of Cycle-SDE reconstructs this trajectory through the analyticly computed $\{w_{t_i}\}_{i=2}^{n+1}$ (see Appendix A.2). However, practical constraints like floating-point precision can lead to reconstruction errors. These errors are especially significant when employing a larger CFG scale, because of the numerical instability inherent to CFG (Lu et al., 2022c). While the reconstruction potential of Cycle-SDE has been touched upon in earlier research (Wu & De la Torre, 2022; Huberman-Spiegelglas et al., 2023), we present a complete discussion here.

Firstly, we presented a visual representation of Cycle-SDE's performance under various CFG scale and machine precision in Fig. 4. Cycle-SDE can flawlessly reconstruct the original image without CFG (i.e., CFG scale 1). However, as we increase the CFG scale, such as to 4, the numerical instability intensifies, thereby hindering the ability to reconstruct the image. We demonstrate that executing experiments with double precision ensures stability and promotes successful image reconstruction, which shows that the primary source of Cycle-SDE's reconstruction error is numerical instability.

Furthermore, we also conducted quantitative experiments to evaluate the image reconstruction capability of Cycle-SDE and ODE inversion. The reconstruction upper bound was determined using the vector-quantized auto-encoder provided by Stable Diffusion (Rombach et al., 2022), denoted as VQAE. We employ Stable Diffusion 1.5 (Rombach et al., 2022) as our foundational model. Following Mokady et al. (2023), we randomly select 100 image-caption pairs from the COCO validation set for our dataset and use PSNR to measure reconstruction quality. For the CFG, Following Mokady et al. (2023), we use a scale 7.5, which is also the default set in Stable Diffusion. We limited our sampling to 50 steps for simplicity.

As outlined in Table 3, Cycle-SDE cannot successfully reconstruct the origin image when employing CFG. Implementing the experiment with double precision mitigates the numerical instability inherent to CFG, significantly reducing Cycle-SDE's reconstruction error, and bringing it closer to the upper bound. Conversely, the reconstruction error in ODE inversion is mainly due to approximation

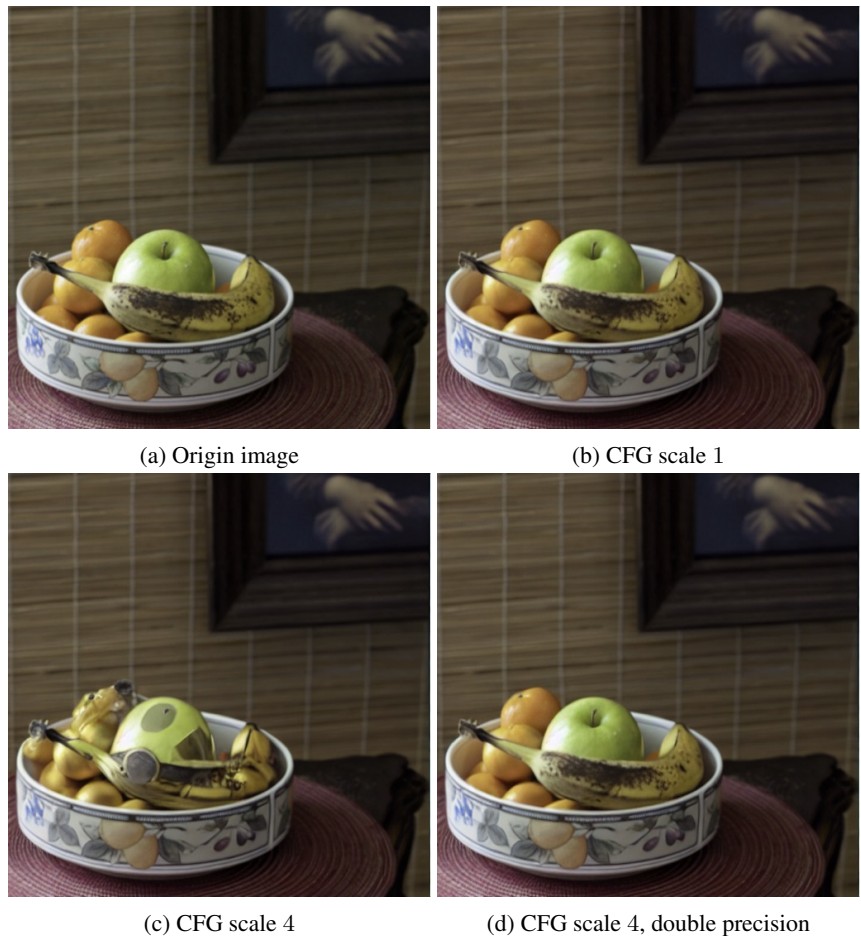

(a) Origin image                      (b) CFG scale 1

(c) CFG scale 4               (d) CFG scale 4, double precision

Figure 4: **Qualitative results in image reconstruction.** Except for (d) which uses double precision, all other reconstruction experiments utilize single precision. Due to numerical instability in CFG, Cycle-SDE struggles to reconstruct the original image while using double precision ensures numerical stability.

Table 3: **Quantitative results in image reconstruction under CFG scale** 7.5. For the experiment of Cycle-SDE, results from three trials were averaged due to randomness. Numerical instability in CFG affects both Cycle-SDE and DDIM inversion in reconstructing the original image. While double precision enhances stability, discretization errors in DDIM still dominate, preventing successful reconstruction.

| | Float32 | | | Float64 | | |
|---|---|---|---|---|---|---|
| | VQAE | DDIM inversion | Cycle-SDE | VQAE | DDIM inversion | Cycle-SDE |
| PSNR | 25.48 | 11.86 | 11.83 | 25.48 | 16.97 | 24.28 |

errors encountered during the ODE discretization phase. Therefore, enhancing machine precision has minimal impact on reducing the reconstruction error of ODE inversion.

## B    DIFFUSION-BASED EDITING METHODS

We present representative diffusion-based editing methods mentioned in Tab. 1 as instances of the general probabilistic formulation in detail.

Inpainting aims to complete missing values based on observation. We start by discussing image sampling and then delve into the distinctions between inpainting and sampling. For a single sampling step in the diffusion model, either an SDE solver or ODE solver is employed to sample from $p_{\boldsymbol{\theta}}(\boldsymbol{x}_s|\boldsymbol{x}_t)$ where $s < t$. In the context of inpainting, SDEdit (Meng et al., 2021) and Stable Diffusion (Rombach et al., 2022) initially generate $\boldsymbol{y}_t$ through the noise-adding process given the reference image $\boldsymbol{y}_0$, and replace the unmasked area of $\boldsymbol{x}_t$ with the unmasked area of $\boldsymbol{y}_t$ to produce $\tilde{\boldsymbol{x}}_t$. SDEdit and Stable Diffusion then sample from $p_{\boldsymbol{\theta}}(\boldsymbol{x}_s|\tilde{\boldsymbol{x}}_t)$ employing an SDE solver and ODE solver, respectively. It is clear that $\tilde{p}_{t_0}(\cdot) \neq p_{t_0}(\cdot)$ due to the manipulation.

DiffEdit (Couairon et al., 2022) and Prompt-to-Prompt (Hertz et al., 2022; Mokady et al., 2023) are methods for image-to-image translation. In the first stage, both methods produce $\boldsymbol{x}_{t_0}$ through ODE inversion employing the source image's prompt $c_1$. For consistency with other methods, we denote $\boldsymbol{x}_{t_0}$ as $\tilde{\boldsymbol{x}}_{t_0}$. In the second stage, they utilize the target prompt $c_2$ for ODE sampling from $\tilde{\boldsymbol{x}}_{t_0}$. Consequently, $\tilde{\boldsymbol{x}}_{t_0} \sim \tilde{p}_{t_0}(\cdot|c_1)$ and a mismatch exists between $\tilde{p}_{t_0}(\cdot|c_1)$ and $p_{t_0}(\cdot|c_2)$.

DDIB (Su et al., 2022) introduces a distinct methodology for image-to-image translation. It employs a probabilistic approach similar to DiffEdit and Prompt-to-Prompt. The primary distinction is that DDIB uses a class-conditional diffusion model, whereas DiffEdit uses a text-conditional diffusion model.

CycleDiffusion (Wu & De la Torre, 2022) is another method for image-to-image translation tasks. CycleDiffusion produce $\boldsymbol{x}_{t_0}$ through the noise-adding process and compute $\bar{\boldsymbol{w}}'_s$ in Eq.(8) employing the source image's label $c_1$. It then utilizes the target label $c_2$ for Cycle-SDE sampling. Similar to DiffEdit and DDIB, $\tilde{p}_{t_0}(\cdot|c_1) \neq p_{t_0}(\cdot|c_2)$ in CycleDiffusion.

DragDiffusion (Shi et al., 2023) and DragonDiffusion (Mou et al., 2023a) are designed for image dragging. In the first stage, both methods produce $\boldsymbol{x}_{t_0}$ through ODE inversion. After optimizing $\boldsymbol{x}_{t_0}$ with gradient descent, they produce $\tilde{\boldsymbol{x}}_{t_0}$ and then employ ODE sampling from $\tilde{\boldsymbol{x}}_{t_0}$. Clearly, we have $\tilde{p}_{t_0}(\cdot) \neq p_{t_0}(\cdot)$ due to the optimization procedure.

# C  RELATED WORK

**Diffusion models.** Diffusion models (Sohl-Dickstein et al., 2015; Ho et al., 2020; Song et al., 2020b) are able to generate high quality imags (Dhariwal & Nichol, 2021), audios (Chen et al., 2020; Kong et al., 2020), videos (Ho et al., 2022; Singer et al., 2022), point clouds (Luo & Hu, 2021), 3D (Poole et al., 2022; Wang et al., 2023), and molecular conformations (Hoogeboom et al., 2022; Bao et al., 2022c). Especially with the emergence of large-scale text-to-image models (Rombach et al., 2022; Ramesh et al., 2022; Saharia et al., 2022; Bao et al., 2023; Balaji et al., 2022; Xue et al., 2023b; Podell et al., 2023), there has been a significant advancement in the domain of image generation.

**SDE and ODE Solvers.** There are some works on increasing the effectiveness of the diffusion model via solving reverse SDE or its equivalent reverse ODE with advanced methods. Ho et al. (2020); Song et al. (2020b); Karras et al. (2022); Lu et al. (2022c); Bao et al. (2022b;a); Jolicoeur-Martineau et al. (2021); Xue et al. (2023a) introduce the reverse SDE discretization methods while Song et al. (2020a); Liu et al. (2022); Lu et al. (2022b;c); Zhang et al. (2022); Karras et al. (2022); Zhao et al. (2023) employ fast ODE sampling algorithm.

In comparison, previous work focuses on sampling from the diffusion model. In contrast, this paper studies SDE and ODE formulations in the context of image editing.

**Diffusion-based image editing method.** Based on the powerful open-set generation capabilities of the large-scale text-to-image diffusion model, the field of image editing has experienced rapid development. Meng et al. (2021); Zhao et al. (2022); Hertz et al. (2022); Wu & De la Torre (2022); Couairon et al. (2022); Bar-Tal et al. (2022); Kawar et al. (2023); Kim et al. (2022); Mokady et al. (2023); Lugmayr et al. (2022); Wang et al. (2022); Chung et al. (2022) introduce advanced inpainting or image-to-image editing method employing various user control. We propose a general probabilistic formulation for image editing and analyze the difference between SDE and ODE when the prior distribution mismatch is due to manipulation or domain transformation during inference. Therefore, our formulation applies to all the above methods. However, training-based methods like Mou et al. (2023b); Zhang et al. (2023) without mismatch, are excluded from our framework.

Inspired by the fast sampling of ODE solvers in image generation, ODE solvers are widely adopted in diffusion-based image editing methods. Our theoretical and empirical evidence suggests that SDE is preferable in editing. Besides, existing SDE-based methods (Meng et al., 2021; Wu & De la Torre, 2022) focus on a specific task but we demonstrate the superiority and versatility of SDE in general image editing.

**Image dragging methods.** Recently, Pan et al. (2023) introduced an interactive Point-based image editing method followed by Shi et al. (2023); Mou et al. (2023a). In comparison, Pan et al. (2023) is a GAN (Goodfellow et al., 2014; Karras et al., 2019) based method, implying its editing capabilities are confined to specific data, such as lions and cats. In contrast, Shi et al. (2023); Mou et al. (2023a) are diffusion-based methods, possessing the capability for open-set editing. However, both of them employ ODE-based formulation, making it possible to further improve.

To our knowledge, SDE-Drag is the first SDE-based method for dragging open-set images. Besides, SDE-Drag manipulates the latent variable in a simple and straightforward copy-and-paste manner instead of performing optimization in the latent space as in all prior work (Pan et al., 2023; Shi et al., 2023; Mou et al., 2023a).

## D    THEORETICAL ANALYSIS

### D.1    THE RATIONALE TO MINIMIZE $KL(\tilde{p}_0||p_0)$

The final objective of image editing is to analyze the editing distribution $\tilde{p}_0$ and data distribution $q_0$ under certain divergence. In this section, we explain why making $\tilde{p}_0$ and $p_0$ close is meaningful both theoretically and empirically.

Theoretically, we indeed assume that the diffusion model characterizes the true score functions, namely $KL(p_0||q_0) = 0$ in Sec. 3.2. Then our theory on $KL(\tilde{p}_0||p_0)$ holds for $KL(\tilde{p}_0||q_0)$ as desired. Intuitively, the assumption can be relaxed to a bounded approximation error of the score function, i.e., $\mathbb{E}_{q_t}[\|s_\theta(\mathbf{x}_t, t) - \nabla_{\mathbf{x}_t} \ln q_t\|^2] < \epsilon$ based on the latest theoretical work (Chen et al., 2022; Lee et al., 2023; Chen et al., 2023). In particular, if $\epsilon \to 0$, then the total variation distance (TV) for $q_0$ and $p_0$ tends to zero, namely, $TV(q_0, p_0) \to 0$, making $p_0$ an accurate approximation for $q_0$.

Empirically, experimental results suggest that $p_0$ is a good surrogate for $q_0$, especially compared to $\tilde{p}_0$. For instance, Stable Diffusion (Rombach et al., 2022) can generate high-fidelity images in human perception. Besides, Tab. 6 shows that the FID for sampling (namely $p_0$) is significantly lower than the FID for editing (ODE baseline, namely $\tilde{p}_0$), suggesting that $p_0$ is much closer to $q_0$ than $\tilde{p}_0$, making it meaningful to minimize $KL(\tilde{p}_0||p_0)$.

### D.2    ASSUMPTIONS

Throughout this section, we adopt the regularity assumptions in Lu et al. (2022a, Assumption A.1). These assumptions are technical and guarantee the existence of the solution to ScoreSDEs, and make the integration by parts and the Fokker-Planck equations valid. For completeness, we list these assumptions in this section.

For simplicity, in the Appendix sections, we use $\nabla(\cdot)$ to denote $\nabla_{\boldsymbol{x}}(\cdot)$ and omit the subscript $\boldsymbol{x}$. And we denote $\nabla \cdot \boldsymbol{h}(\boldsymbol{x}) := \text{tr}(\nabla \boldsymbol{h}(\boldsymbol{x}))$.

**Assumption D.1.** *We make the same assumptions as Lu et al. (2022a, Assumption A.1) and we include them here only for completeness:*

1. *$q_0(\boldsymbol{x}) \in \mathcal{C}^3$ and $\mathbb{E}_{q_0(\boldsymbol{x})}[\|\boldsymbol{x}\|_2^2] < \infty$.*

2. *$\forall t \in [0, T] : \boldsymbol{f}(\cdot, t) \in \mathcal{C}^2$. And $\exists C > 0, \forall \boldsymbol{x} \in \mathbb{R}^d, t \in [0, T] : \|\boldsymbol{f}(\boldsymbol{x}, t)\|_2 \leq C(1 + \|\boldsymbol{x}\|_2)$.*

3. *$\exists C > 0, \forall \boldsymbol{x}, \boldsymbol{y} \in \mathbb{R}^d : \|\boldsymbol{f}(\boldsymbol{x}, t) - \boldsymbol{f}(\boldsymbol{y}, t)\|_2 \leq C\|\boldsymbol{x} - \boldsymbol{y}\|_2$.*

4. *$g \in \mathcal{C}$ and $\forall t \in [0, T], |g(t)| > 0$.*

5. *For any open bounded set $\mathcal{O}$, $\int_0^T \int_{\mathcal{O}} \|q_t(\boldsymbol{x})\|_2^2 + d \cdot g(t)^2 \|\nabla q_t(\boldsymbol{x})\|_2^2 d\boldsymbol{x} dt < \infty$.*

6. $\exists C > 0, \forall \boldsymbol{x} \in \mathbb{R}^d, t \in [0, T] : \|\nabla q_t(\boldsymbol{x})\|_2^2 \leq C(1 + \|\boldsymbol{x}\|_2)$.

7. $\exists C > 0, \forall \boldsymbol{x}, \boldsymbol{y} \in \mathbb{R}^d : \|\nabla \log q_t(\boldsymbol{x}) - \nabla \log q_t(\boldsymbol{y})\|_2 \leq C\|\boldsymbol{x} - \boldsymbol{y}\|_2$.

8. $\exists C > 0, \forall \boldsymbol{x} \in \mathbb{R}^d, t \in [0, T] : \|\boldsymbol{s}_\theta(\boldsymbol{x}, t)\|_2 \leq C(1 + \|\boldsymbol{x}\|_2)$.

9. $\exists C > 0, \forall \boldsymbol{x}, \boldsymbol{y} \in \mathbb{R}^d : \|\boldsymbol{s}_\theta(\boldsymbol{x}, t) - \boldsymbol{s}_\theta(\boldsymbol{y}, t)\|_2 \leq C\|\boldsymbol{x} - \boldsymbol{y}\|_2$.

10. *Novikov's condition:* $\mathbb{E}\left[\exp\left(\frac{1}{2}\int_0^T \|\nabla \log q_t(\boldsymbol{x}) - \boldsymbol{s}_\theta(\boldsymbol{x}, t)\|_2^2 dt\right)\right] < \infty$.

11. $\forall t \in [0, T], \exists k > 0 : q_t(\boldsymbol{x}) = O(e^{-\|\boldsymbol{x}\|_2^k})$, $p_t^{SDE}(\boldsymbol{x}) = O(e^{-\|\boldsymbol{x}\|_2^k})$, $p_t^{ODE}(\boldsymbol{x}) = O(e^{-\|\boldsymbol{x}\|_2^k})$ as $\|\boldsymbol{x}\|_2 \to \infty$.

## D.3 ANALYSES OF SDE AND ODE

**Theorem D.1** (Contraction of SDEs). *Let $p_t$ and $\tilde{p}_t$ be the marginal distributions of two SDEs (see Eq. (4)) at time $t$ respectively. For any $0 \leq s < t \leq T$, if $p_t \neq \tilde{p}_t$, then*

$$D_{\mathrm{KL}}(\tilde{p}_s\|p_s) = D_{\mathrm{KL}}(\tilde{p}_t\|p_t) - \int_s^t g(\tau)^2 D_{\mathrm{Fisher}}(\tilde{p}_\tau\|p_\tau)d\tau < D_{\mathrm{KL}}(\tilde{p}_t\|p_t), \tag{14}$$

*where $D_{\mathrm{KL}}(\cdot\|\cdot)$ denote the KL divergence and $D_{\mathrm{Fisher}}(\cdot\|\cdot)$ denote the Fisher divergence.*

The proof shares the same spirit with previous works (Lyu, 2012; Lu et al., 2022a). In fact, the result is a special case of Proposition C.1. in Lu et al. (2022a). We add proof here for completeness.

*Proof.* We consider a genreal form of $\boldsymbol{f}(\boldsymbol{x}_t, t)$ and $\boldsymbol{f}(\boldsymbol{x}_t, t) = f(t)\boldsymbol{x}_t$ in Eq. (2) is a special case. The two reverse SDEs share the same score model $\theta$ while starting from two different prior distributions $p_t$ and $\tilde{p}_t$ respectively. The process of both reverse SDEs is the same as follows:

$$d\boldsymbol{x}_t = [\boldsymbol{f}(\boldsymbol{x}_t, t) - g(t)^2 \boldsymbol{s}_\theta(\boldsymbol{x}_t, t)]dt + g(t)d\bar{\boldsymbol{w}}_t. \tag{15}$$

However, the induced marginal distributions $p_t$ and $\tilde{p}_t$ are different for any $t \in (0, T]$ because of the different priors, and by the Fokker-Planck equation, we have

$$\frac{\partial p_t(\boldsymbol{x})}{\partial t} = -\nabla_{\boldsymbol{x}} \cdot (\boldsymbol{h}(\boldsymbol{x}, t)p_t(\boldsymbol{x})) \quad \text{and} \quad \frac{\partial \tilde{p}_t(\boldsymbol{x})}{\partial t} = -\nabla_{\boldsymbol{x}} \cdot (\tilde{\boldsymbol{h}}(\boldsymbol{x}, t)\tilde{p}_t(\boldsymbol{x})), \tag{16}$$

where $\nabla_{\boldsymbol{x}}\cdot$ is the divergence operator, and

$$\boldsymbol{h}(\boldsymbol{x}, t) \triangleq \boldsymbol{f}(\boldsymbol{x}, t) - g(t)^2 \boldsymbol{s}_\theta(\boldsymbol{x}, t) + \frac{1}{2}g(t)^2 \nabla_{\boldsymbol{x}} \log p_t(\boldsymbol{x}), \tag{17}$$

$$\tilde{\boldsymbol{h}}(\boldsymbol{x}, t) \triangleq \boldsymbol{f}(\boldsymbol{x}, t) - g(t)^2 \boldsymbol{s}_\theta(\boldsymbol{x}, t) + \frac{1}{2}g(t)^2 \nabla_{\boldsymbol{x}} \log \tilde{p}_t(\boldsymbol{x}), \tag{18}$$

Expanding the time derivative of the KL divergence, we obtain

$$\begin{aligned}
\frac{\partial D_{\mathrm{KL}}(\tilde{p}_t\|p_t)}{\partial t} &= \frac{\partial}{\partial t} \int \tilde{p}_t(\boldsymbol{x}) \log \frac{\tilde{p}_t(\boldsymbol{x})}{p_t(\boldsymbol{x})} d\boldsymbol{x} \\
&= \int \frac{\partial \tilde{p}_t(\boldsymbol{x})}{\partial t} \log \frac{\tilde{p}_t(\boldsymbol{x})}{p_t(\boldsymbol{x})} d\boldsymbol{x} - \int \frac{\tilde{p}_t(\boldsymbol{x})}{p_t(\boldsymbol{x})} \frac{\partial p_t(\boldsymbol{x})}{\partial t} d\boldsymbol{x} \\
&= -\int \nabla_{\boldsymbol{x}} \cdot (\tilde{\boldsymbol{h}}(\boldsymbol{x}, t)\tilde{p}_t(\boldsymbol{x})) \log \frac{\tilde{p}_t(\boldsymbol{x})}{p_t(\boldsymbol{x})} d\boldsymbol{x} + \int \frac{\tilde{p}_t(\boldsymbol{x})}{p_t(\boldsymbol{x})} \nabla_{\boldsymbol{x}} \cdot (\boldsymbol{h}(\boldsymbol{x}, t)p_t(\boldsymbol{x})) d\boldsymbol{x} \\
&= \int (\tilde{\boldsymbol{h}}(\boldsymbol{x}, t)\tilde{p}_t(\boldsymbol{x}))^\top \nabla_{\boldsymbol{x}} \log \frac{\tilde{p}_t(\boldsymbol{x})}{p_t(\boldsymbol{x})} d\boldsymbol{x} - \int (\boldsymbol{h}(\boldsymbol{x}, t)p_t(\boldsymbol{x}))^\top \nabla_{\boldsymbol{x}} \frac{\tilde{p}_t(\boldsymbol{x})}{p_t(\boldsymbol{x})} d\boldsymbol{x} \quad (19) \\
&= \int \tilde{p}_t(\boldsymbol{x})[\tilde{\boldsymbol{h}}(\boldsymbol{x}, t)^\top - \boldsymbol{h}(\boldsymbol{x}, t)^\top][\nabla_{\boldsymbol{x}} \log \tilde{p}_t(\boldsymbol{x}) - \nabla_{\boldsymbol{x}} \log p_t(\boldsymbol{x})] d\boldsymbol{x} \\
&= \frac{1}{2}g(t)^2 \int \tilde{p}_t(\boldsymbol{x})\|\nabla_{\boldsymbol{x}} \log \tilde{p}_t(\boldsymbol{x}) - \nabla_{\boldsymbol{x}} \log p_t(\boldsymbol{x})\|_2^2 d\boldsymbol{x} \\
&= g(t)^2 D_{\mathrm{Fisher}}(\tilde{p}_t\|p_t). \tag{20}
\end{aligned}$$

Eq. (19) holds because of integration by parts with mild regularity assumptions (see Assumption A.1 in Lu et al. (2022a)). Based on Eq. (20), the KL divergence between $p_s$ and $\tilde{p}_s$ is given by:

$$
D_{\mathrm{KL}}(\tilde{p}_s \| p_s) = D_{\mathrm{KL}}(\tilde{p}_t \| p_t) - \int_s^t \frac{\partial D_{\mathrm{KL}}(\tilde{p}_\tau \| p_\tau)}{\partial \tau} d\tau
$$
$$
= D_{\mathrm{KL}}(\tilde{p}_t \| p_t) - \int_s^t g(\tau)^2 D_{\mathrm{Fisher}}(\tilde{p}_\tau \| p_\tau) d\tau
$$
$$
< D_{\mathrm{KL}}(\tilde{p}_t \| p_t).
$$

$\square$

**Theorem D.2** (Invariance of ODEs). *Let $\tilde{p}_t$ and $p_t$ denote the marginal distributions of two ODEs (see Eq. (5) at time $t$ repectively. For any $0 \le s < t \le T$, it holds that*

$$
D_{\mathrm{KL}}(\tilde{p}_s \| p_s) = D_{\mathrm{KL}}(\tilde{p}_t \| p_t). \tag{21}
$$

The proof of Theroem D.2 is similar to Theroem D.1. In particular, it is easy to check that $\frac{\partial D_{\mathrm{KL}}(\tilde{p}_t \| p_t)}{\partial t} = 0$, which is a special case of Theorem 3.1 in Lu et al. (2022a). We add proof here for completeness.

*Proof.* In the ODE setting, $\tilde{\tilde{p}}_t$ and $\tilde{p}_t$ follow the Fokker-Planck equation Eq. (16) with $\boldsymbol{h}(\boldsymbol{x}, t) = \tilde{\boldsymbol{h}}(\boldsymbol{x}, t) = \boldsymbol{f}(\boldsymbol{x}, t) - \frac{1}{2} g(t)^2 \boldsymbol{s}_\theta(\boldsymbol{x}, t)$. In view of Eq. (20), we know that

$$
\frac{\partial D_{\mathrm{KL}}(\tilde{p}_t \| p_t)}{\partial t} = \int \tilde{p}_t(\boldsymbol{x}) [\boldsymbol{h}(\boldsymbol{x}, t)^\top - \tilde{\boldsymbol{h}}(\boldsymbol{x}, t)^\top][\nabla_{\boldsymbol{x}} \log \tilde{p}_t(\boldsymbol{x}) - \nabla_{\boldsymbol{x}} \log p_t(\boldsymbol{x})] d\boldsymbol{x} = 0.
$$

Therefore, it holds that $D_{\mathrm{KL}}(\tilde{p}_s \| p_s) = D_{\mathrm{KL}}(\tilde{p}_t \| p_t)$.

$\square$

## D.4 CONVERGENCE OF SDEs

Following the main text, we have $f(x, t) = -\frac{1}{2} g^2(t) x$ in the following analysis. Below we introduce a widely used functional inequality, and we refer the interested readers to Bakry et al. (2014) for more details. We say a distribution $p$ satisfies the log-Sobolev inequality (LSI) if there exists $c_{\mathrm{LSI}}(p) > 0$ such that the following holds for every distribution $q$:

$$
D_{\mathrm{KL}}(q \| p) \le \frac{c_{\mathrm{LSI}}(p)}{2} \mathbb{E}_q \left\| \nabla \log \frac{q}{p} \right\|^2, \tag{22}
$$

and the smallest constant $c_{\mathrm{LSI}}(p)$ is called the log-Sobolev constant.

**Proposition D.1.** *Suppose that the LSI holds for the data distribution $q_s$ with $c_{\mathrm{LSI}}(q_s) \ge 1$ and $p_t = q_t$. For any $0 \le s < t \le T$, if $p_t \ne \tilde{p}_t$, then*

$$
D_{\mathrm{KL}}(\tilde{p}_s \| p_s) \le \exp\left( -\frac{1}{c_{\mathrm{LSI}}(q_s)} \int_s^t g^2(\tau) d\tau \right) D_{\mathrm{KL}}(\tilde{p}_t \| p_t). \tag{23}
$$

*Proof.* By Lee et al. (2022, Lemma E.7), we see that LSI also holds for $q_\tau$ with $\tau \in [s, t]$ and $c_{\mathrm{LSI}}(q_\tau) \le \max\{c_{\mathrm{LSI}}(q_s), 1\} \le c_{\mathrm{LSI}}(q_s)$. From Eq. (20) and the PI, we know

$$
\frac{d}{d\tau} D_{\mathrm{KL}}(\tilde{p}_\tau \| p_\tau) = g^2(\tau) \mathbb{E}_{\tilde{p}_\tau} \left\| \nabla \log\left( \frac{\tilde{p}_\tau}{p_\tau} \right) \right\|^2 \ge \frac{g^2(\tau)}{c_{\mathrm{LSI}}(q_s)} D_{\mathrm{KL}}(\tilde{p}_\tau \| p_\tau),
$$

which implies that

$$
\frac{d}{d\tau} \left( e^{c_{\mathrm{LSI}}^{-1}(q_s) \int_\tau^t g^2(u) du} D_{\mathrm{KL}}(\tilde{p}_\tau \| p_\tau) \right) \ge 0.
$$

Therefore, the conclusion follows by integrating over $\tau$.

$\square$

### D.5 ANALYSIS OF CYCLE-SDE

We first present a useful lemma.

**Lemma D.1.** *Assume that for any arbitrary values of $\boldsymbol{x}$, $p(\boldsymbol{x}) > 0$, $q(\boldsymbol{x}) > 0$, $q(\boldsymbol{x}|\boldsymbol{y}) > 0$ amd $q(\boldsymbol{x}|\boldsymbol{y}) > 0$. If $p(\boldsymbol{x}|\boldsymbol{y}) = q(\boldsymbol{x}|\boldsymbol{y})$ and $p(\boldsymbol{y}|\boldsymbol{x}) = q(\boldsymbol{y}|\boldsymbol{x})$ holds for all $\boldsymbol{x}$ and for almost every $\boldsymbol{y}$, we have $p(\boldsymbol{x}) = q(\boldsymbol{x})$ for all $\boldsymbol{x}$.*

*Proof.* By the definition of conditional distribution, for all $\boldsymbol{x}$ and almost every $\boldsymbol{y}$, we have

$$\frac{p(\boldsymbol{x},\boldsymbol{y})}{p(\boldsymbol{y})} = \frac{q(\boldsymbol{x},\boldsymbol{y})}{q(\boldsymbol{y})} \quad \text{and} \quad \frac{p(\boldsymbol{x},\boldsymbol{y})}{p(\boldsymbol{x})} = \frac{q(\boldsymbol{x},\boldsymbol{y})}{q(\boldsymbol{x})},$$

which implies that

$$p(\boldsymbol{x})q(\boldsymbol{y}) = q(\boldsymbol{x})p(\boldsymbol{y}),$$

We finish the proof by taking integral w.r.t. $\boldsymbol{y}$ on both sides

$$p(\boldsymbol{x}) = \int p(\boldsymbol{x})q(\boldsymbol{y})d\boldsymbol{y} = \int q(\boldsymbol{x})p(\boldsymbol{y})d\boldsymbol{y} = q(\boldsymbol{x}).$$

$\square$

The SDE inversion algorithm does not follow the SamplingSDE process. However, it still reduces the KL divergence between two marginal distributions induced by two processes from different prior distributions as the time approaches zero. This is formally characterized by the following theorem.

**Theorem D.3** (Contraction of Cycle-SDEs)**.** *Let $p_t$ and $\tilde{p}_t$ be the marginal distributions of two Cycle-SDEs (e.g., see Eq. (6) with $\eta = 0$ and $\bar{\boldsymbol{w}}_s'$ from Eq. (8)) at time $t$ respectively. For any $0 \le s < t \le T$, if $p_t \neq \tilde{p}_t$, then*

$$D_{\mathrm{KL}}(\tilde{p}_s \| p_s) < D_{\mathrm{KL}}(\tilde{p}_t \| p_t). \tag{24}$$

*Proof.* We first introduce two joint distributions of $\boldsymbol{x}_t$ and $\boldsymbol{x}_s$ as follows:

$$p_{t,s}(\boldsymbol{x}_t, \boldsymbol{x}_s) = p_{s|t}(\boldsymbol{x}_s|\boldsymbol{x}_t)p_t(\boldsymbol{x}_t) \quad \text{and} \quad \tilde{p}_{t,s}(\boldsymbol{x}_t, \boldsymbol{x}_s) = p_{s|t}(\boldsymbol{x}_s|\boldsymbol{x}_t)\tilde{p}_t(\boldsymbol{x}_t).$$

where the conditional distribution $p_{s|t}(\boldsymbol{x}_s|\boldsymbol{x}_t)$ is defined by Eq. (6) with $\bar{\boldsymbol{w}}_s$ obtained from Algorithm 5. We denote the conditional distributions by $p_{t|s} = p_{t,s}/p_s$ and $\tilde{p}_{t|s} = \tilde{p}_{t,s}/p_s$.

Then, according to the chain rule for KL divergence Cover (1999), we have that:

$$D_{\mathrm{KL}}(\tilde{p}_{t,s} \| p_{t,s}) = D_{\mathrm{KL}}\left(\tilde{p}_t \parallel p_t\right) + \underbrace{\mathbb{E}_{p_t}\left[D_{\mathrm{KL}}\left(\tilde{p}_{s|t} \parallel p_{s|t}\right)\right]}_{=0}$$

$$= D_{\mathrm{KL}}\left(\tilde{p}_s \parallel p_s\right) + \underbrace{\mathbb{E}_{p_s}\left[D_{\mathrm{KL}}\left(\tilde{p}_{t|s} \parallel p_{t|s}\right)\right]}_{\ge 0},$$

which implies that $D_{\mathrm{KL}}(\tilde{p}_s \| p_s) \le D_{\mathrm{KL}}(\tilde{p}_t \| p_t)$. This is also known as the data processing inequality of KL divergence (Cover, 1999).

The remaining part of the proof shows that $\mathbb{E}_{p_s}\left[D_{\mathrm{KL}}\left(\tilde{p}_{t|s} \parallel p_{t|s}\right)\right] > 0$ by contradiction. In fact, if $\mathbb{E}_{p_s}\left[D_{\mathrm{KL}}\left(\tilde{p}_{t|s} \parallel p_{t|s}\right)\right] = 0$, then $p_{t|s}(\cdot|\boldsymbol{x}_s) = \tilde{p}_{t|s}(\cdot|\boldsymbol{x}_s)$ holds almost surely, and according to Lemma D.1, we have $p_t = \tilde{p}_t$, which is a contradiction. The assumptions in Lemma D.1 hold because of the added Gaussian noise in Algorithm 5. $\square$

## E TOY EXAMPLE

In this section, we conduct a toy simulation on the Gaussian mixture data where the true score function has a closed form and the log-Sobolev inequality holds to illustrate our theoretical results (Theorem 3.1, Theorem 3.2 and Proposition D.1) clearer.

**Data distribution.** We generate examples from a binary mixture of Gaussian distributions, that is, $q_0(x_0) = \frac{1}{2}\mathcal{N}(x_0| - \mu, \sigma^2) + \frac{1}{2}\mathcal{N}(x_0|\mu, \sigma^2)$, where $\mu = 0.5$ and $\sigma = 0.2$. Its score function has

an analytic form, we derive it in the following for completeness. We know that $x_t = \sqrt{\alpha_t}x_0 + \sqrt{1 - \alpha_t}\epsilon_t$, where $\epsilon_t \sim \mathcal{N}(0, 1)$. Then, by changing of variable of probability density, we have

$$\sqrt{\alpha_t}x_0 \sim \frac{1}{2}\mathcal{N}(-\sqrt{\alpha_t}\mu, \alpha_t\sigma^2) + \frac{1}{2}\mathcal{N}(\sqrt{\alpha_t}\mu, \alpha_t\sigma^2),$$

$$\sqrt{1 - \alpha_t}\epsilon_t \sim \mathcal{N}(0, 1 - \alpha_t).$$

Combining these, we have

$$q_t(x_t) = \frac{1}{2}\mathcal{N}(x_t| - \sqrt{\alpha_t}\mu, \alpha_t\sigma^2 + 1 - \alpha_t) + \frac{1}{2}\mathcal{N}(x_t|\sqrt{\alpha_t}\mu, \alpha_t\sigma^2 + 1 - \alpha_t)$$

$$\triangleq \frac{1}{2}\mathcal{N}(x_t| - \mu_t, \sigma_t^2) + \frac{1}{2}\mathcal{N}(x_t|\mu_t, \sigma_t^2)$$

$$= \frac{1}{2\sqrt{2\pi\sigma_t^2}}\exp(-\frac{(x_t - \mu_t)^2}{2\sigma_t^2}) + \frac{1}{2\sqrt{2\pi\sigma_t^2}}\exp(-\frac{(x_t + \mu_t)^2}{2\sigma_t^2}).$$

Therefore, we can obtain the true score function as follows:

$$s(x_t, t) = \nabla_{x_t}\log q_t(x_t)$$

$$= -\frac{1}{q_t(x_t)}\left(\frac{1}{2\sqrt{2\pi\sigma_t^2}}\exp(-\frac{(x_t - \mu_t)^2}{2\sigma_t^2})\frac{x_t - \mu_t}{\sigma_t^2} + \frac{1}{2\sqrt{2\pi\sigma_t^2}}\exp(-\frac{(x_t + \mu_t)^2}{2\sigma_t^2})\frac{x_t + \mu_t}{\sigma_t^2}\right)$$

$$= -\frac{\exp(-\frac{(x_t - \mu_t)^2}{2\sigma_t^2})\frac{x_t - \mu_t}{\sigma_t^2} + \exp(-\frac{(x_t + \mu_t)^2}{2\sigma_t^2})\frac{x_t + \mu_t}{\sigma_t^2}}{\exp(-\frac{(x_t - \mu_t)^2}{2\sigma_t^2}) + \exp(-\frac{(x_t + \mu_t)^2}{2\sigma_t^2})}$$

$$= \frac{\mu_t}{\sigma_t^2}\frac{\exp(-\frac{(x_t - \mu_t)^2}{2\sigma_t^2}) - \exp(-\frac{(x_t + \mu_t)^2}{2\sigma_t^2})}{\exp(-\frac{(x_t - \mu_t)^2}{2\sigma_t^2}) + \exp(-\frac{(x_t + \mu_t)^2}{2\sigma_t^2})} - \frac{x_t}{\sigma_t^2}$$

$$= \frac{\mu_t}{\sigma_t^2}\frac{\exp(\frac{2x_t\mu_t}{\sigma_t^2}) - 1}{\exp(\frac{2x_t\mu_t}{\sigma_t^2}) + 1} - \frac{x_t}{\sigma_t^2}$$

$$= \frac{\mu_t}{\sigma_t^2}\tanh(\frac{\mu_t x_t}{\sigma_t^2}) - \frac{x_t}{\sigma_t^2},$$

where we recall that $\mu_t = \sqrt{\alpha_t}\mu$ and $\sigma_t^2 = \alpha_t\sigma^2 + 1 - \alpha_t$. In our experiment, $\alpha_t$ is set by the cosine schedule in Nichol & Dhariwal (2021).

Besides, the log-Sobolev inequality holds for $q_0$ with $c_{\mathrm{LSI}}(q_0) \leq \sigma\left(1 + \frac{1}{4}\left(e^{\frac{4\mu^2}{\sigma}} + 1\right)\right)$ (see Schlichting (2019, Sec. 4.1) for details).

**Sampler.** Based on the true score function $s_t(x_t)$, we can derive $\epsilon(x_t, t) = -\sqrt{1 - \alpha_t}s(x_t, t)$ and adopt DDPM (i.e., Eq. 6 with $\eta = 1$) as the SDE sampler, DDIM(i.e., Eq. 6 with $\eta = 0$) as the ODE sampler. We sample data from different prior distributions including Gaussian distributions $\mathcal{N}(0, 1)$, $\mathcal{N}(2, 2^2)$ and and uniform distribution $\mathcal{U}[-2, 2]$.

**Results and discussion.** As shown in Fig. 5, when the prior distribution is $\mathcal{N}(0, 1)$, getting benefit from the analytic score function $s(x_t, t)$, both SDE and ODE sampler recover $q_0$. However, when the prior distribution mismatch with the standard Gaussian distribution, the SDE sampler succeeds in recovering $q_0$ while the ODE sampler fails.

The simulation results illustrate our theory (Theorem 3.1, Theorem 3.2 and Proposition D.1) more clearly. On the one hand, Theorem 3.1 and Proposition D.1 state that $D_{\mathrm{KL}}(\tilde{p}_0\|p_0) \approx D_{\mathrm{KL}}(\tilde{p}_0\|q_0)$ is exponentially smaller than $D_{\mathrm{KL}}(\tilde{p}_T\|p_T)$, which means that though the prior mismatch, the SDE sampler can find $\tilde{p}_0$ that is similar to $q_0$. However, in terms of the ODE sampler, Theorem 3.2 guarantees that $D_{\mathrm{KL}}(\tilde{p}_t\|p_t)$ remains unchanged during the sampling process, which means that the ODE sample can never find $\tilde{p}_0$ that is similar to $q_0$.

# F  EXPERIMENTAL DETAILS

We present experimental details in this section.

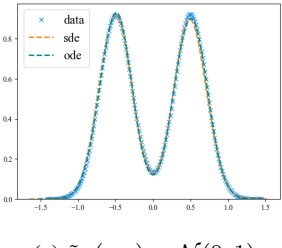 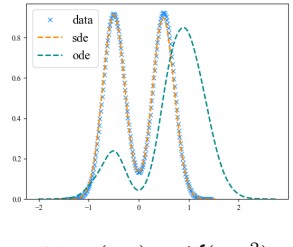 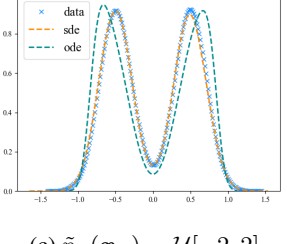

(a) $\tilde{p}_T(\boldsymbol{x}_T) = \mathcal{N}(0,1)$     (b) $\tilde{p}_T(\boldsymbol{x}_T) = \mathcal{N}(2, 2^2)$     (c) $\tilde{p}_T(\boldsymbol{x}_T) = \mathcal{U}[-2, 2]$

Figure 5: **1D toy experiment.** (a) Both ODE and SDE samplers match the data if $\tilde{p}_T(\boldsymbol{x}_T) = \mathcal{N}(0,1)$. (b-c) ODE fails to recover the data distribution while SDE succeeds though the prior distribution mismatch with $\tilde{p}_T(\boldsymbol{x}_T) \neq \mathcal{N}(0,1)$.

### F.1 INPAINTING

We first provide an overview of the inpainting tasks and then detail the experimental settings of Section 5.1.

In the realm of image generation, ODEs converge more rapidly than SDEs (Song et al., 2020b;a; Lu et al., 2022b;c; Karras et al., 2022). As a result, ODEs have been more prevalently utilized in inpainting tasks compared to SDEs. To our knowledge, we are the first to conduct a systematic comparison between SDEs and ODEs in the context of inpainting, demonstrating that SDE can achieve superior results in inpainting tasks.

Inpainting-SDE and inpainting-ODE are equivalent to the inpainting methods used by SDEdit and Stable Diffusion respectively, which are described in Appendix B.

We adopt Stable Diffusion 1.5 as our foundational model and consistently use the prompt "photograph of a beautiful empty scene, highest quality settings" for all images following Rombach et al. (2022). For classifier-free guidance (Ho & Salimans, 2022), we utilize a commonly accepted scale of 7.5. In addition, we employ the mask provided by Li et al. (2022) in evaluation for simplicity.

### F.2 DDIB

In this section, we summarize DDIB (Su et al., 2022) and detail the experimental settings of Appendix G.2.

DDIB is an image-to-image translation method that leverages the reversibility of ODE. Please refer to Appendix B for more details about DDIB. To validate the superiority of SDE in general image editing methods, we replaced the ODE inversion and ODE solver in DDIB with the noise-adding and Cycle-SDE processes, respectively.

For our evaluations, we utilize the FID to measure the similarity between translated images and the target dataset, in addition to the SSIM and L2 metrics to gauge the resemblance between translated images and the source dataset. We set the scale of the classifier guidance to 1 (Dhariwal & Nichol, 2021), consistent with Su et al. (2022).

### F.3 DIFFEDIT

In this section, we give an overview of DiffEdit (Couairon et al., 2022) and provide detailed experimental configurations of Section 5.2.

DiffEdit is a method designed for image-to-image translation. Given a source image $\boldsymbol{x}_0$ accompanied by a source prompt that describes it, and a target prompt that outlines the desired editing image, DiffEdit first generates a mask $M$ to highlight the editing area. Subsequently, the method uses the ODE inversion with the source prompt to obtain the latent representation $\boldsymbol{x}_{t_0}$, where $t_0 \in (0, T]$ serves as a hyper-parameter. DiffEdit then employs the ODE sampling from $\boldsymbol{x}_{t_0}$ employing the target prompt and the mask $M$ to perform inpainting. For the general probabilistic formulation of

Table 4: **Categories in DragBench with corresponding image counts**

| Image type | art | animal | plant | natural landscape | AI-generated |
|:---:|:---:|:---:|:---:|:---:|:---:|
| Count | 5 | 50 | 8 | 20 | 17 |

DiffEdit, please refer to Appendix B. In our SDE implementation, both the ODE inversion and the ODE solver are substituted by the noise-adding and the Cycle-SDE process, respectively.

We adopt the evaluation pipeline from Couairon et al. (2022). For each image in the COCO validation set, COCO-BISON identifies captions that are not directly paired but are similar to the original. Consequently, we possess a source image with an accompanying source prompt and a target prompt paired with a reference image for each edit. We utilize the Stable Diffusion 1.5 (Rombach et al., 2022) as our base model and apply the default classifier-free guidance scale of 7.5 for both DiffEdit-SDE and DiffEdit-ODE. The interval $[0, T]$ is discretized into 100 steps with varying $t_0$. For example, with $t_0 = 0.5T$, both the inversion and sampling procedures consist of 50 steps each.

### F.4 DRAGBENCH

In this section, we present DragBench, our newly introduced benchmark for image dragging. DragBench consists of an open set of 100 images, spanning 5 distinct categories, as detailed in Table 4. Each image in the set is accompanied by a textual description and at least one pair of source-target points. Furthermore, out of the entire collection, 44 images were randomly chosen. For these selected images, masks were applied to regions that do not include the source-target points and visually should remain unchanged. We release DragBench on our project page.

### F.5 USER STUDY

We provide more details about the user study of Section 5.3. Fig. 6 presents a screenshot of the interface in the user study, which includes the guidelines and a sample question provided to the participants.

### F.6 DRAGGING

In this section, we provide a comprehensive overview of our proposed SDE-Drag and outline the implementation details of DragDiffusion and DragGAN.

We employ Stable Diffusion 1.5 (Rombach et al., 2022) as the base model for both SDE-Drag and ODE-Drag. To enjoy relatively high classifier-free guidance (CFG) and numerical stability simultaneously, we linearly increase the CFG scale from 1 to 3 as time goes from 0 to $t_0$.

Subsequently, we delve into the impact of LoRA fine-tuning. As depicted in Figs. (8a-8c), LoRA fine-tuning aids in retaining the core content of the image, particularly in arduous dragging tasks such as revealing the unseen facet of an object. Nonetheless, LoRA fine-tuning may result in overfitting to the input image, compromising its editability, as evidenced in Figs. (8e-8g). To strike a balance between preserving the core content and avoiding overfitting, we adopted a dynamic LoRA scale strategy. Specifically, we reduce the LoRA scale from 1 to 0.5 as time goes from 0 to $t_0$. Note that a LoRA scale of 0 indicates reliance solely on the pre-trained U-Net weights, excluding LoRA parameters. Conversely, at a LoRA scale of 1, all LoRA parameters are utilized. At higher values of $t$ (e.g., $t = t_0$), the denoising process determines the object's outline. Thus, to circumvent overfitting, we opt for a smaller LoRA scale. On the other hand, at lower $t$ values (e.g., $t = 0$), the object's outline is already defined, and the denoising process merely augments details. Therefore, a larger LoRA scale at this stage doesn't pose a risk of overfitting. We assessed the impact of reducing the LoRA scale from 1 to various values $\{0.7, 0.5, 0.3, 0\}$ across several images. Our observations revealed minimal overall differences, leading us to settle on 0.5 as our choice. The influence of the dynamic LoRA scale is evident in Figs. (8d, 8h).

We set the fine-tuning learning rate at $2 \times 10^{-4}$, LoRA rank to 4, and training steps to 100. On a single A100 GPU, this takes about 20 seconds for a single image.

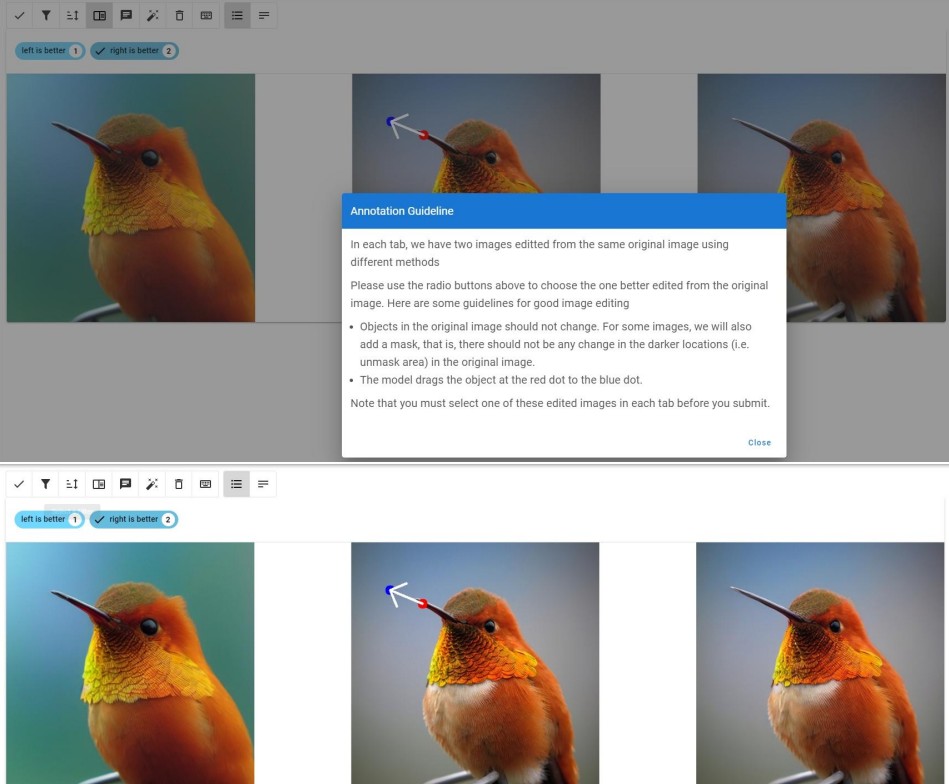

Figure 6: **Screenshot from the User Study.** The top and bottom row displays the guidelines and a sample question provided to the participants, respectively.

For the baseline implementation, we follow the official code provided by DragGAN (Pan et al., 2023) and DragDiffusion (Shi et al., 2023) to edit the images in DragBench for user study and time comparison. For DragDiffusion, we stop optimization when all the source points are no more than $1$ pixel away from the corresponding target points and the maximum number of optimization steps is set to $40$ following the official code. In addition, following DragGAN, we stop optimization when all the source points are no more than $d$ pixel away from the corresponding target points. When the number of point pairs is no more than $5$, $d$ is set to $1$, otherwise it is set to $2$. The maximum number of optimization steps of DragGAN is set to $300$.

## G  ADDITIONAL RESULTS

### G.1  ADDITIONAL RESULTS OF INPAINTING

In this section, to comprehensively demonstrate the superiority of SDE, we present inpainting results using the second-order ODE and SDE solvers. As shown in Tab 5, the SDE algorithm continues to outperform the ODE approach in the context of second-order solvers across all settings. However, when juxtaposed with the first-order solvers, the performance of the second-order algorithms falls short. We conjecture that this underperformance primarily stems from pixel manipulation magnifying the well-known instability tied to high-order algorithms (Lu et al., 2022c).

### G.2  ADDITIONAL RESULTS OF IMAGE-TO-IMAGE TRANSLATION

In this section, we present the results of another image-to-image translation method named DDIB (Su et al., 2022), illustrating the superiority of SDE compared to the ODE baseline over the general image editing methods.

Table 5: **Results in inpainting under second order solver.** Inpainting-ODE-2 and Inpainting-SDE-2 employ DPM-Solver++(2M) and SDE-DPM-Solver++(2M) (Lu et al., 2022c) respectively. Inpaint-SDE-2 outperforms Inpaint-ODE-2 in all settings.

| | Small Mask | | | | | | Large Mask | | | | | |
|---|---|---|---|---|---|---|---|---|---|---|---|---|
| | FID ↓ | | | LPIPS ↓ | | | FID ↓ | | | LPIPS ↓ | | |
| # steps | 25 | 50 | 100 | 25 | 50 | 100 | 25 | 50 | 100 | 25 | 50 | 100 |
| Inpainting-ODE-2 | 6.12 | 6.03 | 5.95 | 0.229 | 0.228 | 0.227 | 18.13 | 17.82 | 17.67 | 0.363 | 0.362 | 0.361 |
| Inpainting-SDE-2 | **6.11** | **5.39** | **5.10** | **0.224** | **0.219** | **0.217** | **18.04** | **16.20** | **15.61** | **0.355** | **0.347** | **0.344** |

Table 6: **Results in DDIB.** Editing and Sampling represent image-to-image translation and image generation, respectively. SDE significantly outperforms ODE under FID in image-to-image translation and exhibits a more consistent performance between editing and sampling.

| | Lion → Tiger | | | Cock → Bird | | |
|---|---|---|---|---|---|---|
| | FID ↓ | l2 ↓ | SSIM ↑ | FID ↓ | l2 ↓ | SSIM ↑ |
| *Editing* | | | | | | |
| DDIB-ODE | 30.25 | 68.21 | 0.059 | 58.02 | 83.63 | 0.135 |
| DDIB-SDE | 16.55 | 67.83 | 0.063 | 26.63 | 87.13 | 0.160 |
| *Sampling* | | | | | | |
| ODE | 15.71 | - | - | 29.51 | - | - |
| SDE | 17.56 | - | - | 28.62 | - | - |

**Setup.** For fairness and consistency, we employ the ImageNet dataset and ADM (Dhariwal & Nichol, 2021) trained on ImageNet in the evaluation following DDIB. We adopt the wide-use metrics FID, L2 and SSIM (Wang et al., 2004) in image-to-image translation (Meng et al., 2021; Zhao et al., 2022; Wu & De la Torre, 2022) for quantitative analysis. We keep all hyperparameters the same as DDIB for simplicity and fairness. Furthermore, in order to visually illustrate the difference between editing and sampling, we also conduct image-generation experiments on the target dataset, which is a sampling from Gaussian noise that employs the target dataset label. For more discussion about the DDIB experiment please refer to Appendix F.2.

**Results.** As presented in Tab. 6, for translations from Lion to Tiger and Cock to Bird, the DDIB-SDE (i.e., the SDE-based DDIB) significantly outperforms the DDIB-ODE (i.e., the ODE-based DDIB) in image fidelity, as gauged by the FID metric. While achieving high image fidelity, DDIB-SDE preserves similar image faithfulness to DDIB-ODE as measured by the L2 and SSIM metrics. Please refer to Fig. 13 for visualization results, in which DDIB-SDE shows a higher image fidelity in image-to-image translation.

For ODE, there is a significant performance difference between editing and sampling, with the editing FID being notably inferior. Conversely, SDE demonstrates more uniform performance, with the editing and sampling FID scores closely aligned. This observation robustly validates our motivation that SDE achieves superior consistency between editing and sampling.

### G.3 ADDITIONAL RESULTS OF USER STUDY

In this section, we present additional user study results of SDE-Drag over ODE-Drag and DragGAN without LoRA. As shown in Fig. 7, SDE-Drag consistently outperforms the two baselines.

### G.4 ADDITIONAL RESULTS OF TIME EFFICIENCY

In this section, we discuss the time efficiency. As shown in Table 7, in all experiments, the SDE counterpart takes nearly the same time as the direct ODE baseline.

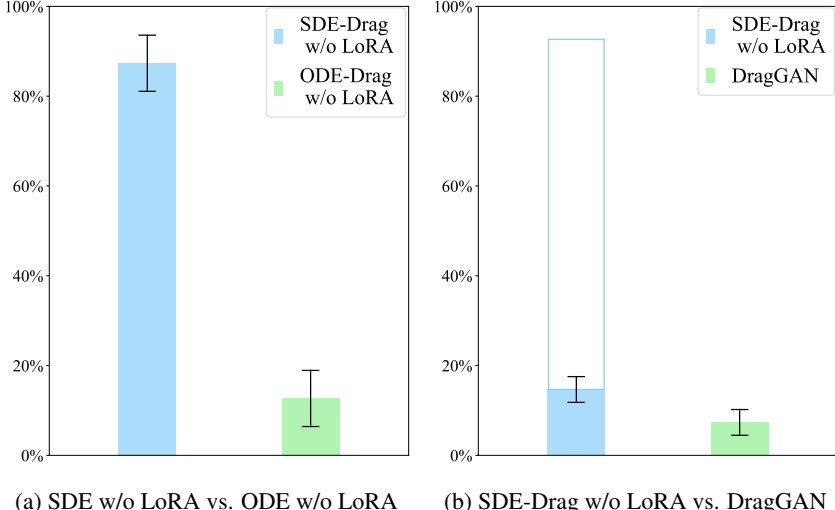

(a) SDE w/o LoRA vs. ODE w/o LoRA    (b) SDE-Drag w/o LoRA vs. DragGAN

Figure 7: **Results of dragging without LoRA.** We present the preference rates (with 95% confidence intervals) of SDE-Drag over ODE-Drag and DragGAN without LoRA. SDE-Drag significantly outperforms all competitors. The blank box in (b) denotes the ratio of the open-domain images in DragBench that DragGAN cannot edit.

Table 7: **Additional results in per-image time costs.** The SDE counterpart takes nearly the same time as the direct ODE baseline.

|        | Inpainting    | DiffEdit      | DDIB                    | Dragging      |
|--------|---------------|---------------|-------------------------|---------------|
| SDE    | 3.06s         | 4.82s         | 299.20s                 | 75.86s        |
| ODE    | 3.11s         | 4.86s         | 300.18s                 | 76.62s        |
| Device | NVIDIA A100   | NVIDIA A100   | NVIDIA GeForce RTX 3090 | NVIDIA A100   |

## H  HYPERPARAMETER ANALYSIS OF SDE-DRAG

We present more sensitivity analysis results of SDE-Drag in this section. Notably, SDE-Drag is not sensitive to most of the hyperparameters. To highlight the roles of individual parameters in SDE-Drag, we showcase images that respond differently to specific hyperparameter adjustments.

Fig. 8 shows the effect of finetuning LoRA. In addition, we display the analysis of hyperparameter $t_0$, $\alpha$, $\beta$ and $m$ in Fig. 9, Fig. 10, Fig. 11 and Fig. 12, respectively.

## I  VISUALIZATION RESULTS

### I.1  DETAILS OF DDIB

As presented by Fig. 13, DDIB-SDE shows a higher image fidelity in image-to-image translation compared with DDIB-ODE.

### I.2  DETAILS OF DIFFEDIT

In Table 8, we present the quantitative metrics from the DiffEdit experiment, corresponding to Fig. 2 in Section 5.2. Additionally, Fig. 14 offers a visual comparison between DiffEdit-SDE and DiffEdit-ODE, demonstrating that DiffEdit-SDE ensures superior image quality and better image-text alignment.

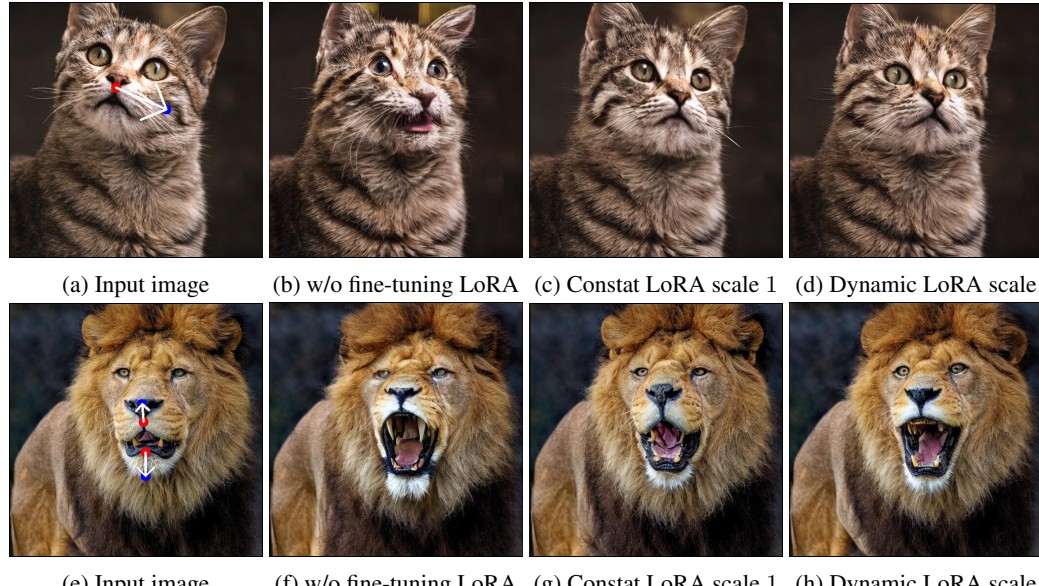

| (a) Input image | (b) w/o fine-tuning LoRA | (c) Constat LoRA scale 1 | (d) Dynamic LoRA scale |
| (e) Input image | (f) w/o fine-tuning LoRA | (g) Constat LoRA scale 1 | (h) Dynamic LoRA scale |

Figure 8: **Analysis of LoRA fine-tuning**. (a-c) demonstrate that LoRA fine-tuning aids in preserving object consistency during editing, whereas (e-g) indicates a potential risk of overfitting to the input image. (d, h) illustrate how the dynamic LoRA scale strategy, as detailed in Appendix F.6, effectively balances between content retention and overfitting prevention.

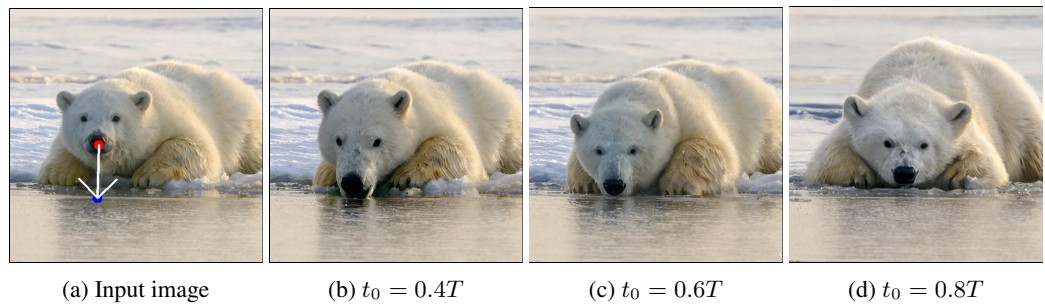

| (a) Input image | (b) $t_0 = 0.4T$ | (c) $t_0 = 0.6T$ | (d) $t_0 = 0.8T$ |

Figure 9: **Analysis of hyper-parameter** $t_0$. A lower $t_0$ will lead to greater faithfulness to the input image but may compromise image quality. Conversely, a higher $t_0$ enhances image fidelity at the expense of faithfulness. However, in most of the cases, SDE-Drag is robust to $t_0$ between $0.5T$ and $0.7$ T so we set the default $t_0$ as $0.6$ T.

## I.3 DRAG COMPARE

In this section, we provide visual comparisons of SDE-Drag with DragDiffusion and DragGAN in Fig. 15 and Fig. 16, respectively.

## I.4 DRAG FUN

In this section, we showcase the editing outcomes of SDE-Drag. The edits of real images are illustrated in Fig. 17. Edits of images generated by Stable Diffusion (Rombach et al., 2022; Podell et al., 2023) are shown in Fig. 18, and edits of DALL·E 3 generated images can be seen in Fig. 19. Moreover, we highlight the methodology of dragging multiple points sequentially in Fig. 20.

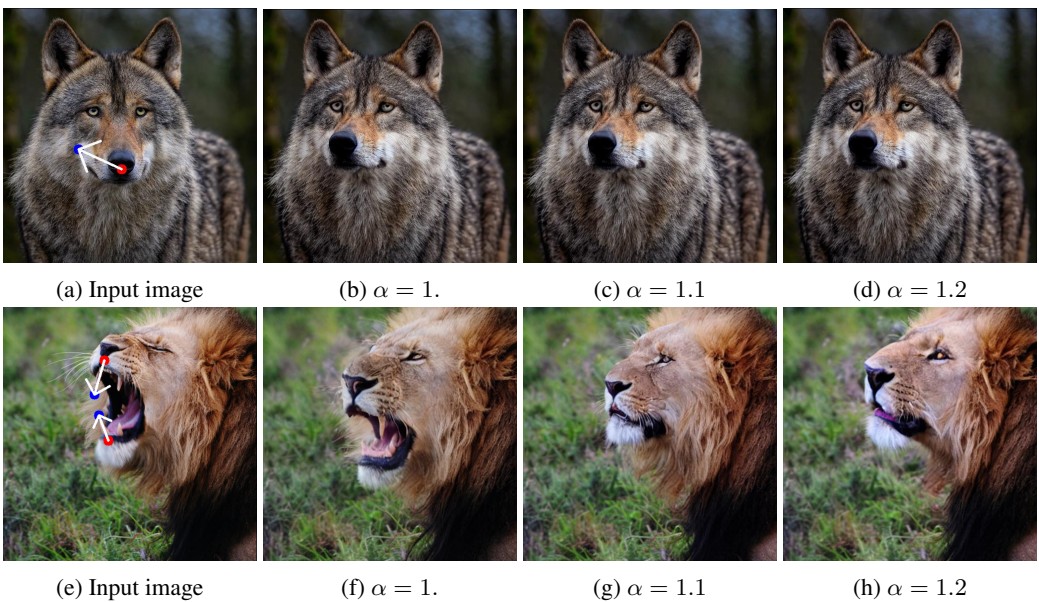

(a) Input image     (b) $\alpha = 1$.     (c) $\alpha = 1.1$     (d) $\alpha = 1.2$

(e) Input image     (f) $\alpha = 1$.     (g) $\alpha = 1.1$     (h) $\alpha = 1.2$

Figure 10: **Analysis of hyper-parameter** $\alpha$. SDE-Drag is not sensitive to the value of $\alpha$ in most of the cases as shown in the top row. However, we observed that a small number of images achieve better editing results when $\alpha = 1.1$ as evidenced in bottom line. Therefore, we set the default $\alpha$ to 1.1.

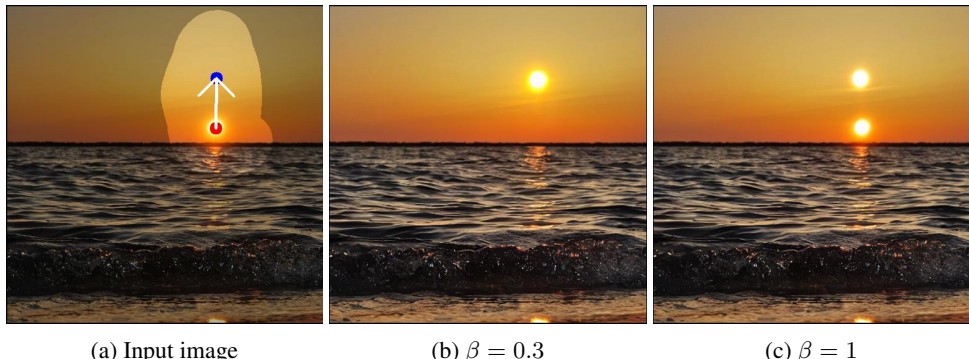

(a) Input image        (b) $\beta = 0.3$        (c) $\beta = 1$

Figure 11: **Analysis of hyper-parameter** $\beta$. For clarity, we fix the hyper-parameter $m$ to 1 in this figure. When $\beta = 1$ the object at the source point is retained, which can be used for "copying" objects. A $\beta$ value between 0.1 and 0.5 is suitable for "dragging" and produces nearly identical editing outcomes. Consequently, we set the default $\beta$ as 0.3.

## J   FAILURE CASE

We show a failure case of SDE-Drag in Fig. 21 which indicates that effectively dragging open-set images is still a significant challenge.

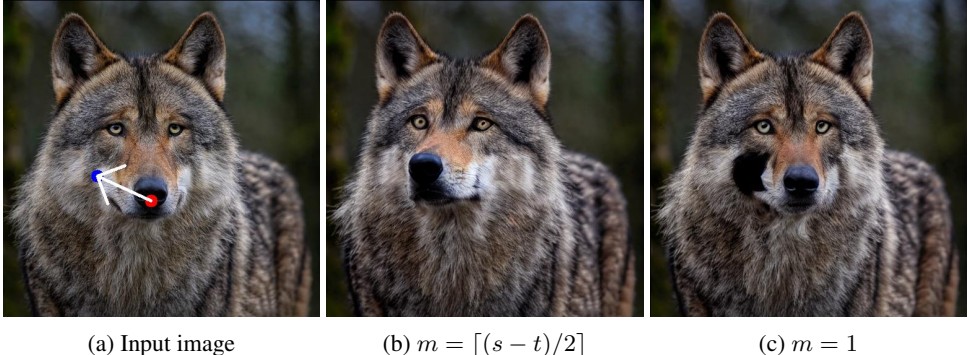

|  (a) Input image  |  (b) $m = \lceil (s-t)/2 \rceil$  |  (c) $m = 1$  |

Figure 12: **Analysis of hyper-parameter** $m$. $m = \lceil \|a_s - a_t\|/2 \rceil$ represent that the distance in each dargging operation is 2 pixels. We found that 1 pixel and 4 pixels per operation also work well. However, it is challenging when $m = 1$.

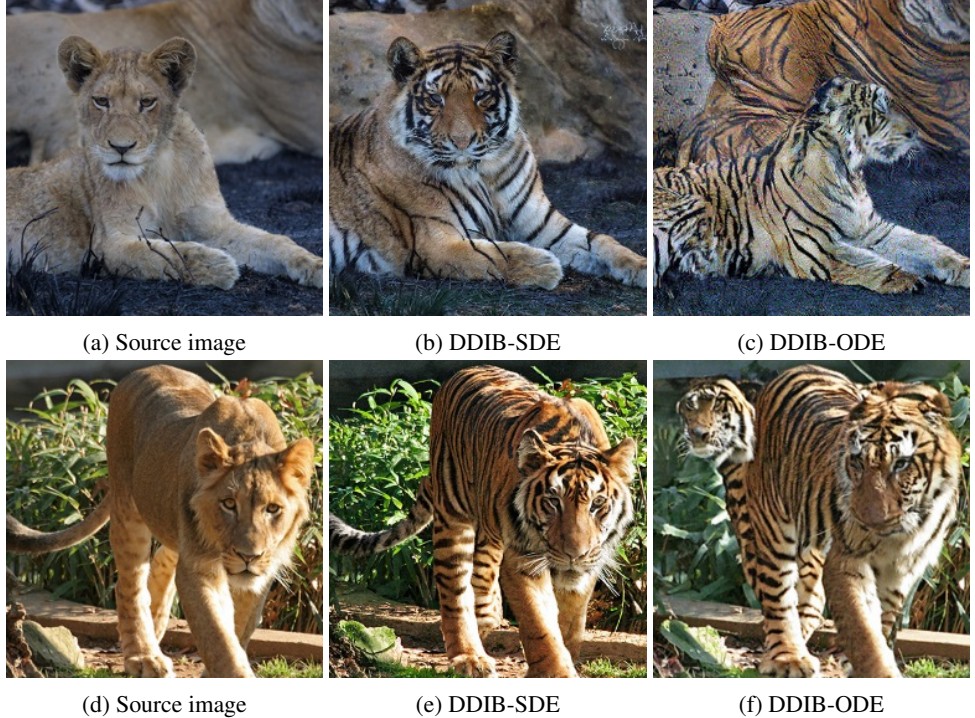

|  (a) Source image  |  (b) DDIB-SDE  |  (c) DDIB-ODE  |
|  (d) Source image  |  (e) DDIB-SDE  |  (f) DDIB-ODE  |

Figure 13: **The visualization results of DDIB.** The source class is "lion", the target class is "tiger". DDIB-SDE achieves significantly better image fidelity compared to DDIB-ODE.

Table 8: **Quantitative results of DiffEdit**. For all evaluation metrics, DiffEdit-SDE consistently surpasses DiffEdit-ODE across different $t_0$.

| Model | FID $\downarrow$ | Clip-Score $\uparrow$ | LPIPS $\downarrow$ |
|---|---|---|---|
| $t_0 = 0.3$ | | | |
| DiffEdit-SDE | **7.20** | **0.219** | **0.171** |
| DiffEdit-ODE | 7.67 | 0.215 | 0.173 |
| $t_0 = 0.4$ | | | |
| DiffEdit-SDE | **7.29** | **0.224** | **0.181** |
| DiffEdit-ODE | 8.24 | 0.215 | 0.184 |
| $t_0 = 0.5$ | | | |
| DiffEdit-SDE | **7.57** | **0.228** | **0.195** |
| DiffEdit-ODE | 9.42 | 0.215 | 0.199 |
| $t_0 = 0.6$ | | | |
| DiffEdit-SDE | **7.97** | **0.230** | **0.209** |
| DiffEdit-ODE | 11.27 | 0.214 | 0.214 |
| $t_0 = 0.7$ | | | |
| DiffEdit-SDE | **8.65** | **0.232** | **0.219** |
| DiffEdit-ODE | 13.51 | 0.212 | 0.228 |
| $t_0 = 0.8$ | | | |
| DiffEdit-SDE | **9.25** | **0.232** | **0.227** |
| DiffEdit-ODE | 15.91 | 0.210 | 0.240 |

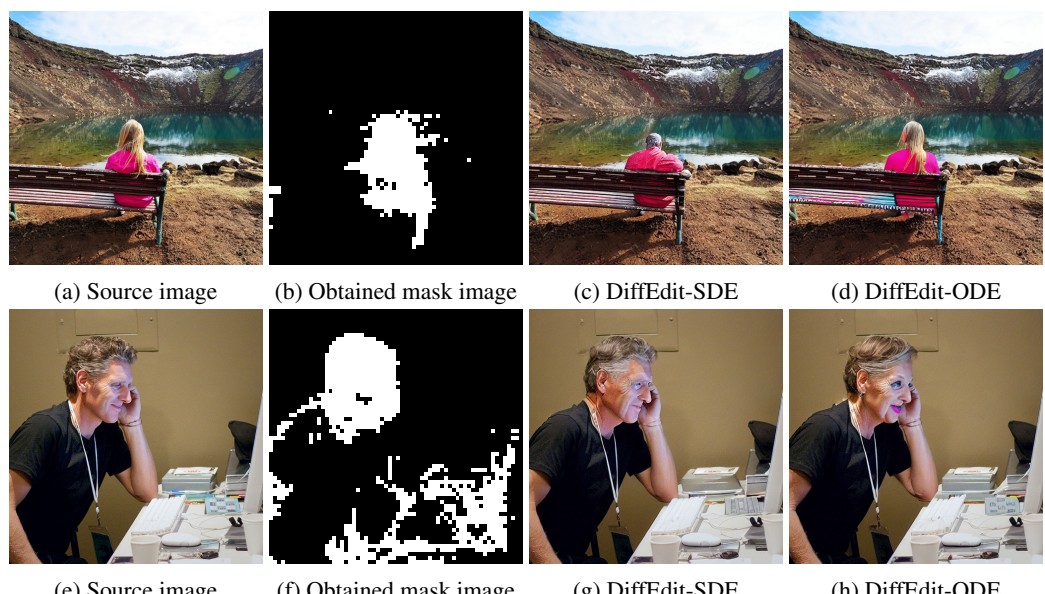

(a) Source image  (b) Obtained mask image  (c) DiffEdit-SDE  (d) DiffEdit-ODE

(e) Source image  (f) Obtained mask image  (g) DiffEdit-SDE  (h) DiffEdit-ODE

Figure 14: **Visualization results of Diffedit.** For the top row, the source prompt is "`Young female sitting on a bench near a small stagnant lake`" and the target prompt is ""`A man sitting on a wooden bench near a lake.`" For the bottom row, the source prompt is "`A man sitting down at a table using a computer.`", and the target prompt is "`An older man sitting at a desk looking at a laptop.`". DiffEdit-SDE achieves significantly better image fidelity and image-text alignment compared to DiffEdit-ODE.

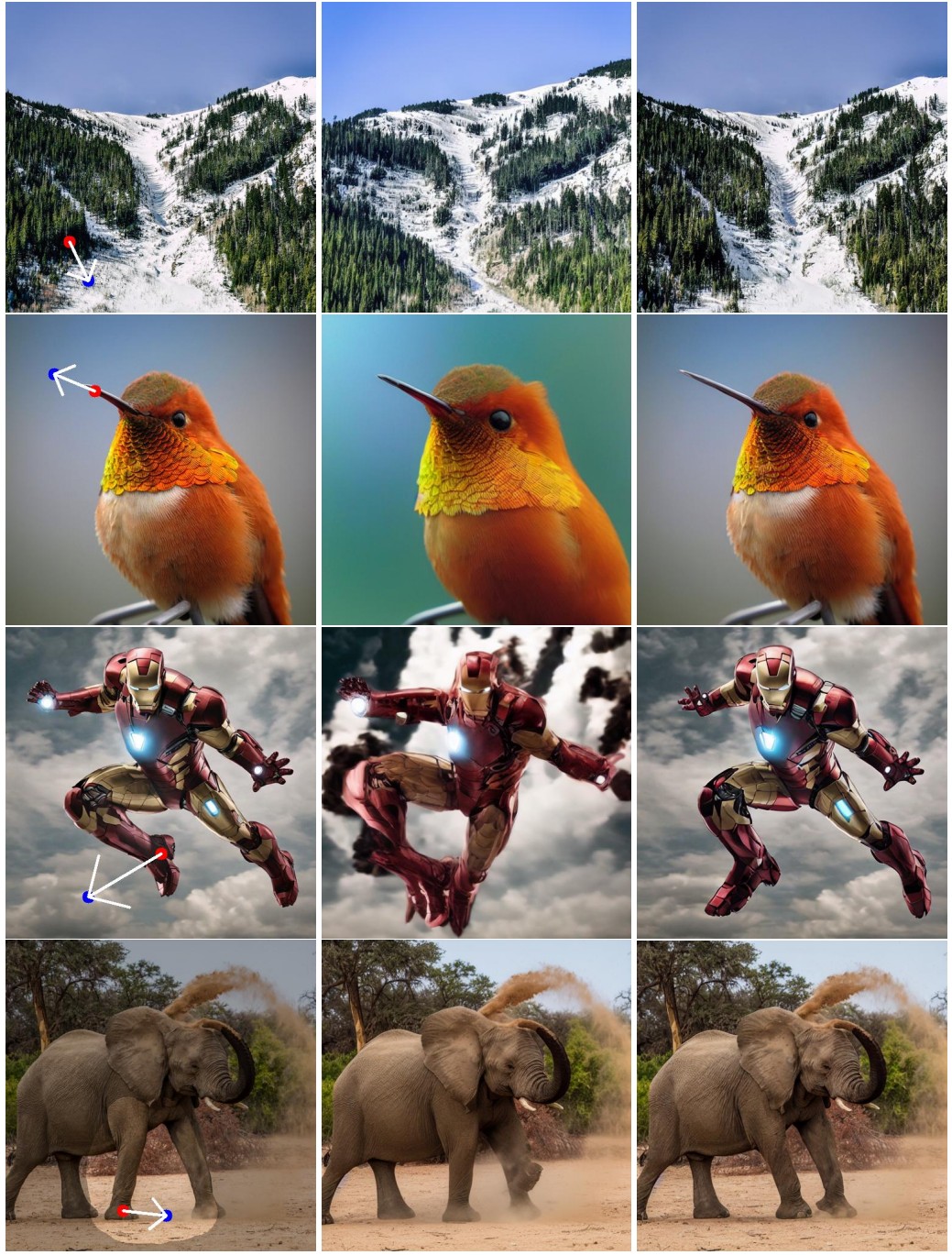

Figure 15: **Visual comparison between DragDiffusion and SDE-Drag.** The left column displays the original input images, while the center and right columns present edits from DragDiffusion and SDE-Drag, respectively.

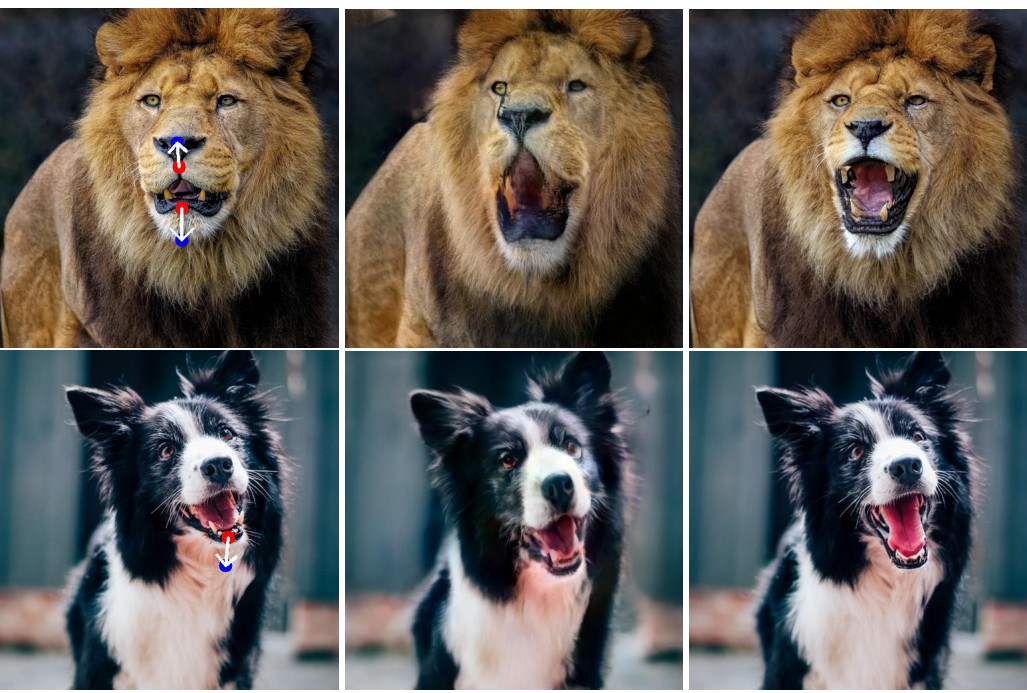

Figure 16: **Visual comparison between DragGAN and SDE-Drag.** The left column displays the original input images, while the center and right columns present edits from DragGAN and SDE-Drag, respectively.

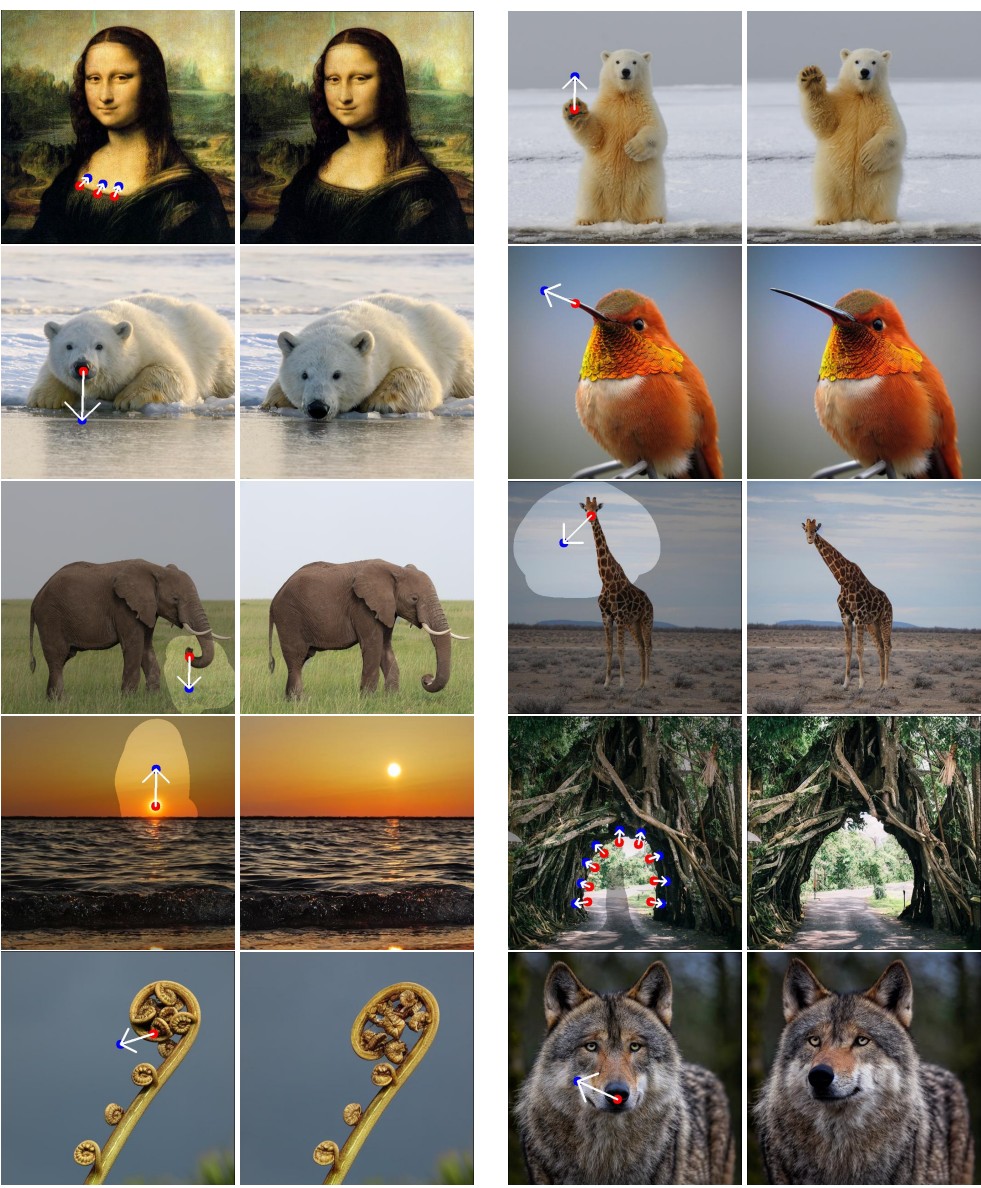

Figure 17: **Editing results of SDE-Drag in real images.**

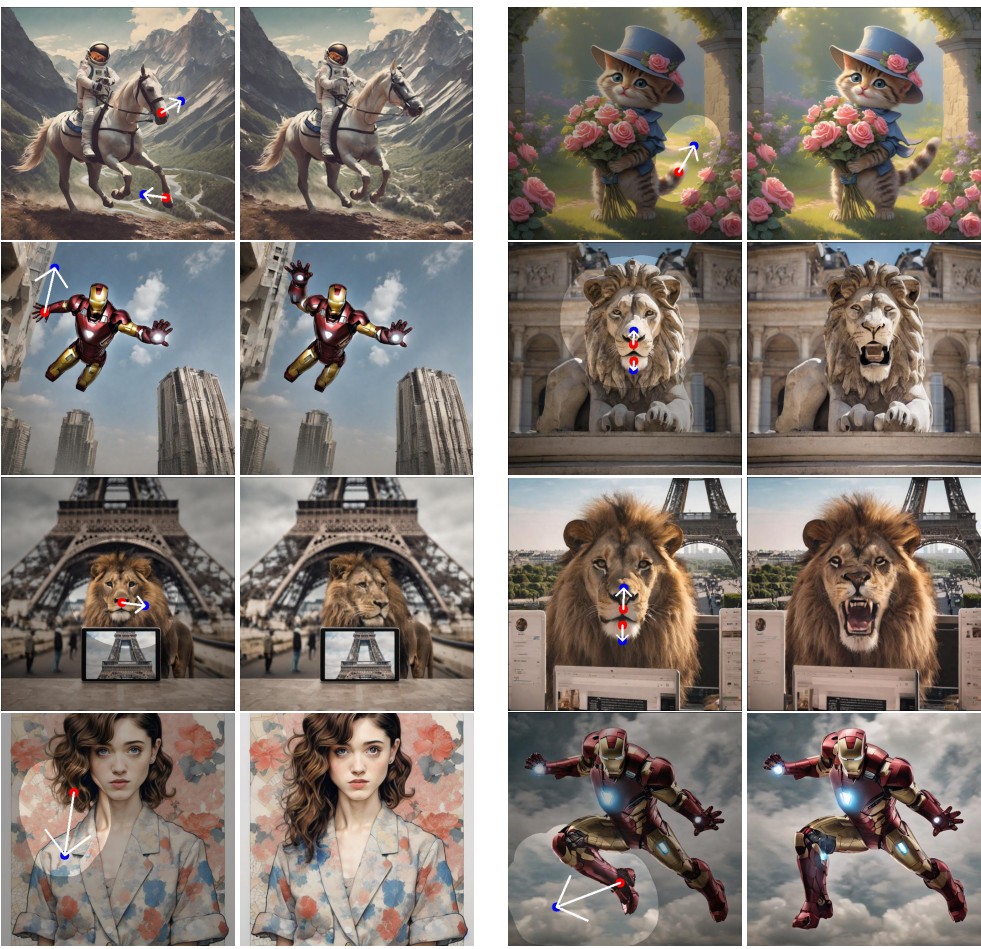

Figure 18: **Editing results of SDE-Drag in images generated by Stable Diffusion.**

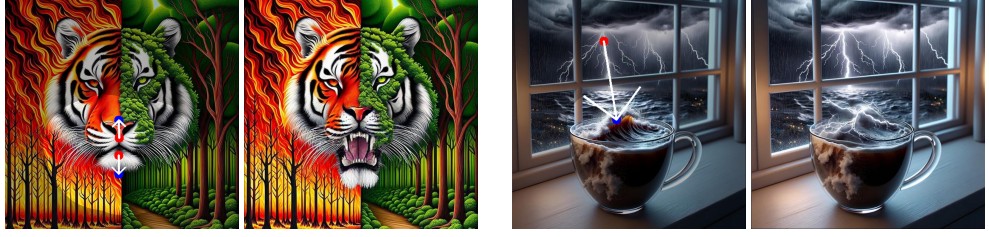

Figure 19: **Editing results of SDE-Drag in images generated by DALL·E 3.**

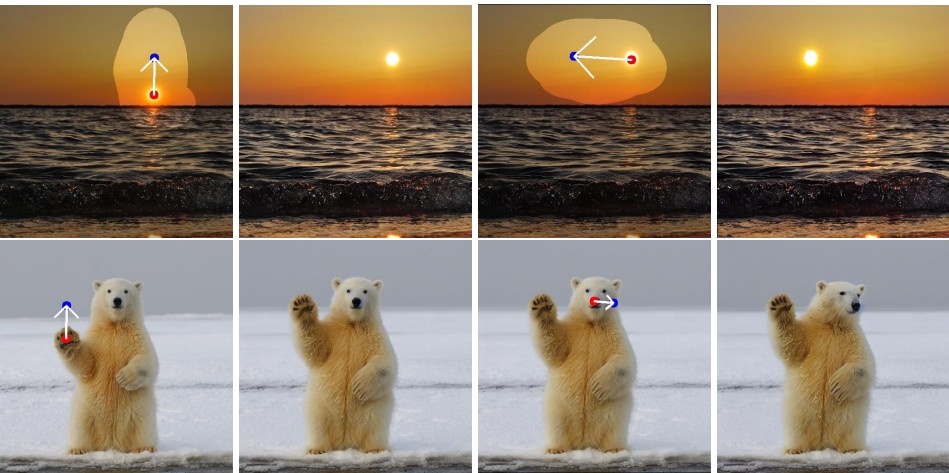

Figure 20: **Results of dragging multiple points sequentially.**

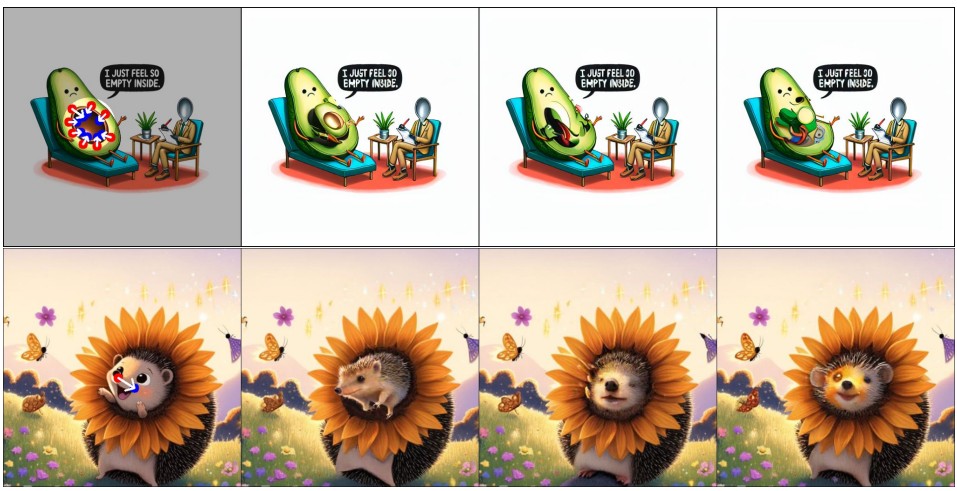

Figure 21: **Failure case of SDE-Drag.** For each input image shown on the far left, we tested SDE-Drag with three different random seeds. Such results suggest that dragging open-set images is still a significant challenge.

