# OpenReview forum: "The Blessing of Randomness: SDE Beats ODE in General Diffusion-based Image Editing"
_ICLR.cc/2024/Conference — ICLR 2024 poster_

### Official Review · Reviewer_mBr1 · 2023-10-19

**Soundness:** 3 good
**Presentation:** 2 fair
**Contribution:** 3 good
**Rating:** 8
**Confidence:** 4

**Summary:**

The paper presents a unified probabilistic formulation for diffusion-based image editing and introduces a simple yet effective dragging algorithm based on this formulation. The authors conduct experiments on various tasks, including inpainting, image-to-image translation, and dragging, demonstrating the superiority of their SDE-based approach.

**Strengths:**

1. The paper provides a comprehensive theoretical analysis of the SDE and ODE formulations for general image editing.
2. The authors propose the SDE-Drag algorithm for dragging.
3. The experiments across different tasks, including inpainting, image-to-image translation, and dragging, demonstrate the effectiveness of the SDE formulation in improving image editing tasks compared with the ODE baselines.
4. The authors also provide the code, which shows the solidness of the work.

**Weaknesses:**

1. While the paper shows that SDE outperforms ODE baselines in inpainting and image-to-image translation, it lacks a comparison with the latest methods in these tasks.
2. The paper provides the cost time for the dragging task but does not provide similar information for inpainting and image-to-image translation.

**Questions:**

1. It is suggested to compare the performance of SDE in inpainting and image-to-image translation with the latest methods.
2. Provide the running time of their proposed methods for inpainting and image-to-image translation tasks. And including the running time of the latest methods in these tasks to provide a more complete picture of the computational efficiency.

---

> ### Author Response · Authors · 2023-11-17
> **Response to Reviewer mBr1**
>
> We thank reviewer mBr1 for the acknowledgment of our contributions and the valuable comments. We respond below to your questions and concerns.
>
> ## Weakness1: Comparison with the latest methods.
>
>
> Thank you for your valuable comment. The preliminary goal of the inpainting and image-to-image translation experiments is to show the versatility of SDE in general image editing, instead of improving the latest methods. Our dragging experiment against the latest baselines is able to show the effectiveness of the SDE formulation. Besides, based on the results, we believe that the SDE formulation is orthogonal to other improvements for a specific task and applies to the latest ODE-based methods.
>
> Moreover, in our submission, we conduct experiments on recent work including SD (CVPR 2022), DiffEdit (ICLR 2023), DDIB (ICLR 2023), and DragDiffusion (arXiv 2023.06). Nevertheless, if you would like to point out specific papers, we are happy to discuss the relation and will try to do the experiments.
>
>
> ## Weakness2: Costing time for inpainting and image-to-image translation tasks.
> Thank you for your valuable question. In the last paragraph (on page 9) of Sec. 5.4 in our submission, we introduce that the SDE counterpart takes nearly the same time as the direct ODE baseline in all experiments. To more clearly demonstrate the time efficiency, we list the cost time per image for all inpainting and image-to-image translation experiments here and we will add it to the revision as soon as possible.
>
>
> |       | Inpainting | DiffEdit   | DDIB                   |
> |-------|------------|------------|------------------------|
> | SDE   | 3.06s      | 4.82s      | 299.20s                |
> | ODE   | 3.11s      | 4.86s      | 300.18s                |
> | Device| NVIDIA A100 | NVIDIA A100 | NVIDIA GeForce RTX 3090|
>
>
>
> From the table, we can see that the time efficiency of both SDE and ODE mainly depends on the number of function evaluations, which are the same using the same sampling strategy. Therefore, when applied to the latest ODE baselines, SDE-based methods would have a similar time cost. Also see the discussion on comparison with the latest methods in our response to your weakness1.

---

> > ### Comment · Reviewer_mBr1 · 2023-11-22
> > **After rebuttal**
> >
> > Thank you for your rebuttal. The author's explanation regarding the completeness of the comparison is satisfactory to me. Additionally, they provide detailed cost-time comparisons. Overall, the author has addressed my concerns. Therefore, I would like to maintain my rating as borderline accept.

---

> > > ### Author Response · Authors · 2023-11-22
> > >
> > > Thank you for your feedback.

---

### Official Review · Reviewer_UXKo · 2023-10-25

**Soundness:** 3 good
**Presentation:** 3 good
**Contribution:** 3 good
**Rating:** 6
**Confidence:** 4

**Summary:**

This paper focuses on image editing using pre-trained diffusion models. The authors propose a unified probabilistic formulation for diffusion-based image editing, including inpainting, image-to-image translation, and dragging, based on existing methods. Experiments show that the proposed method achieves better performance than the original ODE version.

**Strengths:**

1. The authors provide SDE versions for existing ODE-based editing methods and achieve better performance.

2. The authors build a benchmark called DragBench for evaluation, which may benefit the community.

3. The paper is well written, and the experiment and presentation are solid.

**Weaknesses:**

1. The core idea that SDE beats ODE in image editing seems similar to CycleDiffusion [6]. This article seems to just generalize this phenomenon to multiple tasks and verify them. Overall, the innovative contribution seems insufficient.

2. The authors claim to propose a unified probabilistic formulation for diffusion-based image editing. However, many related image editing/I2I methods are not mentioned, e.g., RePaint [1], DDNM [2], DPS [3], T2I-Adapter [4], and ControlNet [5].

References:

[1] Lugmayr et al., Repaint: inpainting using denoising diffusion probabilistic models, CVPR 22

[2] Wang et al., Zero-shot image restoration using denoising diffusion null-space model, ICLR 23

[3] Chung et al., Diffusion posterior sampling for general noisy inverse problems, ICLR 23

[4] Mou et al., T2I-adapter: learning adapters to dig out more controllable ability for text-to-image diffusion models, arXiv 23

[5] Zhang et al., Adding conditional control to text-to-image diffusion models, ICCV 23

[6] Wu et al., A latent space of stochastic diffusion models for zero-shot image editing and guidance, ICCV 23

**Questions:**

Please see the Weaknesses.

---

> ### Author Response · Authors · 2023-11-17
> **Response to Reviewer UXKo**
>
> We thank reviewer UXKo for the acknowledgment of our contributions and the valuable comments. We respond below to your questions and concerns.
>
> ## Weakness1: The relationship between our work and CycleDiffusion.
>
> Thank you for your valuable question. we emphasize our contributions and explain the differences and links between our work and CycleDiffusion in detail:
>
> 1. We introduce a unified perspective for multiple diffusion-based image editing tasks (I2I, inpainting, dragging) that encompasses various editing techniques. In contrast, CycleDiffusion focuses on I2I.  **Given the fact that existing research in such tasks is mainly independent, we believe the unified perspective is a nontrivial and valuable contribution.**
> 2. **To our knowledge, we present the FIRST theoretical analysis (which is one of our core contributions) on the superiority of SDE over ODE in image editing, while all existing methods including Cycle-Diffusion are purely empirical.**
> 3. We present a new algorithm (SDE-Drag) as well as a new benchmark for open-set image dragging. Besides the SDE formulation, **the core contribution of SDE-Drag is a simple copy&paste manipulation strategy for the latent variables**, which significantly outperforms existing optimization-based methods. Such contributions are unique in our paper compared to existing work including Cycle-Diffusion.
> 4. **The core idea of Cycle-Diffusion is to construct an invertible SDE algorithm**, inspired by the ODE methods. **It is mainly compared with non-invertible SDE baselines**. However, our core idea is that **the general SDE formulation (including Cycle-Diffusion and others) is better than ODE in editing**. Our theory applies to other SDE formulations and we apply other SDE-based methods in the inpainting task other than Cycle-Diffusion.
>
> We believe the above contributions make our paper distinct from Cycle-Diffusion.
>
> ## Weakness2: More image editing methods are not mentioned.
>
> Thank you for your insightful suggestion and for pointing out the related works, as briefly discussed below.
>
> Our framework analyzes the difference between SDE and ODE when the prior distribution mismatch is due to manipulation or domain transformation during inference. Therefore, it also applies to RePaint, DDNM, and DPS. However, training-based methods like T2I-Adapter and ControlNet without mismatch, are excluded from our framework.
>
> We will add the above discussion in our revision.

---

> > ### Comment · Reviewer_UXKo · 2023-11-21
> >
> > The author's rebuttal addresses most of my concerns, and I'll keep my score.

---

> > > ### Author Response · Authors · 2023-11-22
> > >
> > > Thank you for your time and review.

---

### Official Review · Reviewer_NG3Y · 2023-10-27

**Soundness:** 3 good
**Presentation:** 3 good
**Contribution:** 3 good
**Rating:** 5
**Confidence:** 3

**Summary:**

This paper show theoretically and experimentally the benefit of using diffusion model SDEs over ODEs for image editing. The authors formulate the image editing with diffusion model process in three steps : 1) encoding with deterministic or random noise 2) alteration of the latent which means modification of the prior distribution representing this latent 3) SDE or ODE sampling starting from the altered latent.
On the theoretical side, it is proven that during step 3, the KL distribution between the SDE marginals : a) when sampling from the altered latent and b) sampling from the original latent, decreases, while remaining constant when sampling with ODEs. On the experimental side, it is analyzed, from different works, the benefit of using SDEs over ODEs in step 3.

**Strengths:**

The argumentation is limpid, and the contributions are clearly stated.

Although I am not an expert in image editing, I think that the main strength of the paper is its experimental study, which looks impressive. It contains numerous comparisons in three different tasks.  Moreover, the authors created an evaluation benchmark for point dragging and conducted a user study for evaluation on this problem. The advantage of SDEs over ODEs is clearly demonstrated experimentally.

I like the fact that the authors took care to expose a very easy Gaussian toy example to illustrate the theorems.

**Weaknesses:**

Major weaknesses :
- The problematic of the paper (end of Section 3.1) is never clearly answered. Actually, I do not think that the paper properly gives the answer to this question. This is link to the following point.
- It is not clear that the proposed theoretical arguments prove the right point. Given $x_0$ (resp. $\tilde x_0$) sampled from the latent $x_{t_0}$ (resp.  $\tilde x_{t_0}$. ), with Theorem 3.1, it is proven that the $x_0$ and $\tilde x_0$ are closer “in distribution” than $x_{t_0}$ and $\tilde x_{t_0}$. Why is that desired ?
I think that the authors see $p_0$ as the distribution of clean images $q_0$, and then wish to minimize $KL(p_0, \tilde p_0)$ to get well-looking images. However, $p_0$ is very different from $q_0$ because the score is not (and for from being) perfectly matched with the denoiser. From this fact, could the authors explain why it makes sense to try to minimize $KL(p_0, \tilde p_0)$ ? I am more likely to think that, for the purpose of image editing, it does not make sense to try to minimize the distance between these two distributions.
- In the experimental section, some important information is missing : which model are you using, trained on which dataset ?

Minor weaknesses :
- In the paragraph "Samples" from Section 2. The term "equivalent" is not true. Sampling is not "equivalent" to discretization ! Moreover, discretizing (4) or (5) does not enable to sample from $q_0$ if the score is not perfectly matched with $\epsilon_\theta$.
- The ODE inversion process explanation should be clarified. The links and differences between deterministic ODE inversion and random Gaussian noise should be explained.
- The "mild assumption" should be detailed, at least in the Appendix.

**Questions:**

- If the noise is fixed, is CycleSDE still an SDE ?
- Is the log-Sobolev inequality likely to be verified in practice ?

I am prepared to improve my score by taking into account the author's feedback.

---

> ### Author Response · Authors · 2023-11-17
> **Response to Reviewer NG3Y Part1**
>
> We thank reviewer NG3Y for the acknowledgment of our contributions and the valuable comments. We respond below to your questions and concerns.
>
> ## Weakness1: The reason to minimize $KL(\tilde p_0 || p_0)$ .
>
>
> Thank you for your meticulous and insightful comment. We agree that the final objective of image editing is to analyze $\tilde p_0$ and $q_0$ under certain divergence. Below, we explain **why making $\tilde p_0$ and $p_0$ close (i.e., the contribution of this paper) is meaningful both theoretically and empirically**.
>
> Theoretically, we indeed assume that the diffusion model characterizes the true score functions, namely $KL(p_0 || q_0)=0$ $(p_0=q_0)$ in the submission (See Line 3, Paragraph 1 in Sec. 3.2). Then our theory on $KL(\tilde p_0|| p_0)$ holds for $KL(\tilde p_0||q_0)$ as desired. **Intuitively, the assumption can be relaxed to a bounded approximation error of the score function**, i.e., $\mathbb E_{q_t}[ \|s_{\mathbf \theta}(\mathbf x_t, t) - \nabla_{\mathbf x_t} \ln q_t \|^2]  < \epsilon$ based on the latest theoretical work [a, b, c]. In particular, if $\epsilon \rightarrow 0$, then the total variation distance (TV) for $q_0$ and $p_0$ tends to zero, namely, $TV(q_0, p_0) \rightarrow 0$, making $p_0$ an accurate approximation for $q_0$.
>
> Empirically, **experimental results suggest that $p_0$ is a good surrogate for $q_0$, especially compared to $\tilde p_0$**. For instance, Stable Diffusion [d] can generate high fidelity images in human perception. Besides, Tab. 5（on page 26）in our submission shows that **the FID for sampling (namely $p_0$) is significantly lower than the FID for editing (ODE baseline, namely $\tilde p_0$)**, suggesting that $p_0$ is much closer to $q_0$ than $\tilde p_0$, making it meaningful to minimize $KL(\tilde p_0|| p_0)$.
>
> We sincerely appreciate the valuable comments. We will revise the paper to clarify that the final objective of image editing is to analyze $\tilde p_0$ and $q_0$ and present the above discussion on why minimizing $KL(\tilde p_0 || p_0)$ is meaningful. If you have any further concerns, please feel free to post more comments.
>
>
> [a] Chen, Sitan, et al. "Sampling is as easy as learning the score: theory for diffusion models with minimal data assumptions." *arXiv preprint arXiv:2209.11215* (2022).
>
> [b] Lee, Holden, Jianfeng Lu, and Yixin Tan. "Convergence of score-based generative modeling for general data distributions." *International Conference on Algorithmic Learning Theory*. PMLR, 2023.
>
> [c] Chen, Sitan, Giannis Daras, and Alex Dimakis. "Restoration-degradation beyond linear diffusions: A non-asymptotic analysis for ddim-type samplers." International Conference on Machine Learning. PMLR, 2023.
>
> [d] Rombach, Robin, et al. "High-resolution image synthesis with latent diffusion models." Proceedings of the IEEE/CVF conference on computer vision and pattern recognition. 2022.
>
>
> ## Weakness2: Pretrained models and datasets in the experimental section.
>
> Thank you for thoroughly reading our paper and highlighting the issues in detail. **For fairness, we use the same pretrained models (including the training dataset) as the corresponding baselines for all experiments.** In particular, we use Stable Diffusion 1.5 [d] trained on the LAION dataset [e] for the inpainting, DiffEdit and dragging tasks. We use ADM [f] trained on the ImageNet dataset [g] for the DDIB experiment.
>
> We will add the information in Sec. 5 in the revision.
>
> [e] Schuhmann, Christoph, et al. "Laion-5b: An open large-scale dataset for training next generation image-text models." *Advances in Neural Information Processing Systems* 35 (2022): 25278-25294.
>
> [f] Dhariwal, Prafulla, and Alexander Nichol. "Diffusion models beat gans on image synthesis." Advances in neural information processing systems 34 (2021): 8780-8794.
>
> [g] Deng, Jia, et al. "Imagenet: A large-scale hierarchical image database." 2009 IEEE conference on computer vision and pattern recognition. Ieee, 2009.

---

> ### Author Response · Authors · 2023-11-17
> **Response to Reviewer NG3Y Part2**
>
> ## Weakness3: Expression in the paragraph "Samplers" from Section 2.
>
> Thank you very much for pointing out our mistakes. We will change our expression: **Samples can be generated from the diffusion model by discretizing the reverse SDE (Eq.(4)) or ODE (Eq.(5)) from $T$ to $0$**.
>
> ## Weakness4: Details about ODE inversion.
>
> Thank you for your insightful suggestion. We are not sure about the "random Gaussian noise" in your comment. We guess you mean the $\bar{w}_s$ in Eq. (6) in the submission. **Note that when using ODE inversion, we set $\eta=0$, which implies that $\sigma_s = 0$. Namely, the coefficient of the random Gaussian noise in Eq. (6) is zero** and it is a deterministic process. If we misunderstand your comment, please feel free to post further questions.
>
> In case you are not familiar with ODE inversion, it generally aims to obtain $\mathbf x_T$ given $\mathbf x_0$ based on the invertibility of the ODE. According to Eq. (5) in the submission, the general formulation of ODE inversion is defined as $\mathbf x_T = \mathbf x_0 + \int_0^T \left[f(t) \mathbf x_t + \frac{g^2(t)}{2 \sqrt{1 - \alpha_t}}  \mathbf \epsilon_{\mathbf \theta}(\mathbf x_t, t)\right] dt$. This implies that for each given data point $\mathbf x_0$, there exists a uniquely determined $\mathbf x_T$ associated with it through ODE inversion.
>
> As for the specific DDIM sampler, one step is Eq. (6) in our submission. We rewrite it here for clarity (only ODE formulation, i.e., $\eta=0$): $\mathbf x_t = \sqrt{\alpha_{t}} \left( \frac{\mathbf x_s - \sqrt{1 - \alpha_s} \mathbf \epsilon_{\mathbf \theta}(\mathbf x_s, s)}{\sqrt{\alpha_s}} \right) + \sqrt{1 - \alpha_{t}} \mathbf \epsilon_{\mathbf \theta}(\mathbf x_s, s)$. When $t>s$, it is a step of DDIM inversion.
>
> We will provide the details of ODE inversion in the Appendix in the revision.
>
>
> ## Weakness5：Assumption in our theoretical analyses.
>
> Thank you for your insightful suggestion. For completeness, we will list the assumption in Appendix D, which is the same as Assumption A.1. of [h].
>
> [h] Lu, Cheng, et al. "Maximum likelihood training for score-based diffusion odes by high order denoising score matching." *International Conference on Machine Learning*. PMLR, 2022.
>
> ## Question1: Is CycleSDE still an SDE?
>
> Thank you for your question. We interpret "noise is fixed" to mean that the noises in the backward process of Cycle-SDE are solved by the forward process. Actually, given $\mathbf x_{t_0}$, the noises in the backward process are not iid standard Gaussian noise like the origin SDE, but instead a combination of noises in the forward process.  Consequently, the reverse process is stochastic, and thus Cycle-SDE is still an SDE.
>
>
> ## Question2: Is the log-Sobolev inequality hold for the data distribution in practice?
>
> Thank you for your question. The log-Sobolev inequality is a **classical and standard assumption in theoretical analyses of Langevin-type algorithms [i, j], which cannot be verified in large-scale and high-dimensional real data** like LAION [e]. Note that without the assumption, although it is nontrivial to obtain the strong linear convergence result, our main results (i.e., Theorem 1,2&3) still hold.
>
> [i] Chewi, Sinho, et al. "Analysis of Langevin Monte Carlo from Poincar\'e to Log-Sobolev." arXiv preprint arXiv:2112.12662 (2021).
>
> [j] Vempala, Santosh, and Andre Wibisono. "Rapid convergence of the unadjusted Langevin algorithm: Isoperimetry suffices." Advances in neural information processing systems 32 (2019).
>
>
> **Finally, we believe the quality of the paper has been improved following your insightful suggestions. If you have any more questions, we are happy to discuss them and will do our best to address them!**

---

> ### Author Response · Authors · 2023-11-20
> **Looking forward to further feedbacks**
>
> Dear Reviewer NG3Y,
>
> Thank you again for your great efforts and valuable comments. We have carefully addressed the main concerns in detail. We hope you will find the response satisfactory. As the discussion phase is about to close, we are very much looking forward to hearing from you about any further feedback. We will be very happy to clarify further concerns (if any).
>
> Best, Authors

---

> > ### Comment · Reviewer_NG3Y · 2023-11-21
> > **Response to rebuttal**
> >
> > Thanks for your careful rebuttal. The authors answered all my interrogations. I will take a bit more time to think about my score.

---

> ### Author Response · Authors · 2023-11-22
>
> We thank Reviewer NG3Y for the time and consideration. We are happy to see the comment *"The authors answered all my interrogations"* and there is no further question. We hope the Reviewer NG3Y can update the score accordingly based on the initial review *"I am prepared to improve my score by taking into account the author's feedback."*
>
> We understand that Reviewer NG3Y can be very busy but we kindly remind Reviewer NG3Y that the author-reviewer discussion period ends in one day.

---

### Author Response · Authors · 2023-11-20
**Summary of the revision**

We sincerely thank all reviewers for their valuable comments, which help to further improve the quality of our work. We have thoroughly addressed the detailed comments, and summarize the revision in the updated version as follows:

* We add a discussion to clarify that the final objective of image editing is to analyze $\tilde p_0$ and $q_0$ and present the rationale of minimizing $KL(\tilde p_0|| p_0)$ (See Appendix D.1).
* We detail the pre-trained models and the training datasets (See Paragraph "Setup" in Sec. 5.1, Paragraph "Setup" in Sec. 5.2, and Paragraph "Setup" in Appendix G.2)
* We change our wrong expression (See Paragraph "Samplers" in Sec. 2)
* We add more details about the ODE inversion (See Paragraph 2 in Appendix A.2)
* We list the assumptions (See Appendix D.2)
* We present a discussion about more image editing methods (See Paragraph 3 in Appendix C)
* We list a cost-time comparison between SDE and ODE (See Appendix G.4)

We hope you will find the response satisfactory. Please let us know if you have any further feedback.

---

### Meta-Review · Area_Chair_SCih · 2023-12-06

**Metareview:**

This paper presents an invertible SDE diffusion model for image editing that surpasses the quality of existing ODE solvers. The authors  provide a theory for existence and  convergence to solutions. The motivation of their approach, as pointed out by a reviewer is not clear, even though the empirical results is in favor of their hypothesis.

**Justification For Why Not Higher Score:**

The technical novelty is limited to unify existing editing approaches.

**Justification For Why Not Lower Score:**

The theoretical results are significant for image editing performance.

---

### Decision · Program_Chairs · 2024-01-16

Accept (poster)